# ENHANCING CROSS-TASK TRANSFER OF LARGE LANGUAGE MODELS VIA FOURIER ACTIVATION STEERING

## ABSTRACT

Large Language Models (LLMs) have shown impressive abilities in leveraging pretrained knowledge through prompting, but they often struggle with unseen tasks, especially in data-scarce scenarios. While cross-task in-context learning provides a direct solution for knowledge transfer without fine-tuning, it still faces limitations in terms of robustness, scalability, and efficiency. In this paper, we investigate *whether cross-task transfer can be achieved via **latent space steering***. Through analysis of activation patterns under both zero-shot and few-shot prompts, we have three observations: (1) the activation differences between few-shot and zero-shot prompts exhibit a nearly parallel structure in low-dimensional space; (2) these difference vectors correlate strongly with task similarity; (3) Fourier analysis reveals that low-frequency components encode task-agnostic, information-enhanced features, while high-frequency components capture task-specific details. Motivated by these findings, we propose **FAST**, a **F**ourier-based **A**ctivation **S**teering cross-task **T**ransfer framework. It first selects influential and diverse samples from high-resource tasks, then injects information-enhanced low-frequency components along with task-similarity weighted high-frequency components during inference. Extensive experiments in both cross-domain and cross-lingual transfer settings show that our method consistently outperforms existing methods. The code is available in `https://anonymous.4open.science/r/RETL-BC5B`.

## 1 INTRODUCTION

Large Language Models (LLMs) (OpenAI, 2024; Guo et al., 2025) demonstrate remarkable capabilities to store knowledge during pretraining, which can be effectively accessed via prompting. However, as this paradigm becomes increasingly prevalent, a critical challenge emerges: these models often struggle with tasks that were not seen during pretraining, particularly in data-scarce scenarios (Bigoulaeva et al., 2025). A common strategy to tackle this issue is transfer learning (Zhuang et al., 2021; Strangmann et al., 2024; Somerstep et al., 2025), which uses knowledge from high-resource tasks to adapt to low-resource ones. Several studies (Vu et al., 2022; Li et al., 2022; Lv et al., 2024) fine-tune soft prompts on data-sufficient tasks and subsequently apply them to data-scarce tasks during inference. While effective, such approaches still require training and can not generalize well across diverse tasks. An alternative line of work investigates cross-task in-context learning (ICL) (Tanwar et al., 2023; Li et al., 2023b; Chatterjee et al., 2024), which utilizes labeled examples from high-resource tasks to improve performance on low-resource tasks without parameter updates.

Despite its promise, cross-task ICL still faces several limitations. (1) Performance is highly sensitive to the choice of demonstrations (Liu et al., 2022a; Levy et al., 2023), prompt templates (Wang et al., 2023; Mishra et al., 2022), and source tasks (Chatterjee et al., 2024), which restricts its adaptability (**robustness**). (2) Few demonstrations can be included due to the constrained context length of LLMs (**scalability**). (3) Computational cost increases significantly with more demonstrations due to the quadratic complexity of Transformers to the input length (**efficiency**). To address these challenges, one possible solution is to operate activations in the continuous space, which avoids expanding the discrete token sequences and mitigates context length constraints. This leads us to raise a research question: *Can we achieve effective cross-task transfer through **latent space steering**?*

To answer this question, we conduct an empirical study of activation patterns under both zero-shot and few-shot prompts. Our analysis reveals three main findings: (1) The difference in activations between

few-shot and zero-shot prompts exhibits a **nearly parallel structure** when projected into a low-dimensional space across diverse task pairs. This suggests that the enhanced information provided by in-context examples has **consistent directions** in the model's activation space. (2) To explore whether these difference activations can be directly injected, we measure their directional similarity across different source and target tasks in the high-dimensional activation space. We observe that their directions are **highly correlated with task similarity**, suggesting that directly injecting difference activations is more effective for related tasks, and less effective for dissimilar ones. (3) Inspired by Fourier analysis, where representations can be decomposed into broad trends (low-frequency) and localized details (high-frequency), we apply Fourier-based filtering to disentangle the difference vectors. We find that low-frequency components capture **task-agnostic and information-enhanced features**, and high-frequency components contain **task-specific information**.

Building on these insights, we propose a **F**ourier-based **A**ctivation **S**teering cross-task **T**ransfer framework, namely **FAST**. We first select a representative subset of high-resource examples that balance influence and diversity, aiming to improve the efficiency of feature extraction. Specifically, we first construct a similarity graph of high-resource examples, and then iteratively select a sample with the highest combined influence and diversity score at each step until the desired subset size is reached. Next, we compute the activation differences between few-shot and zero-shot prompts for these selected samples. We then apply Fourier-based filtering to decompose these difference activations into information-enhanced features (*i.e.,* low-frequency components) and task-specific features (*i.e.,* high-frequency components). During inference on low-resource queries, we inject the low-frequency components along with task-similarity weighted high-frequency components into the forward pass, which guides LLMs toward effective cross-task transfer. This approach is both efficient and scalable, as it relies solely on pre-computed activations from high-resource tasks without requiring parameter updates or expanded input length. To validate the effectiveness of our proposed method, we conduct extensive experiments in both cross-domain and cross-lingual scenarios. Experimental results demonstrate that **FAST** consistently outperforms competitive baselines while maintaining lower computational cost. Our main contributions can be summarized as follows:

• To the best of our knowledge, we are the first to systematically analyze the activation patterns of both zero-shot and few-shot prompts. We find that: (1) the activation differences between few-shot and zero-shot prompts exhibit a consistent and nearly parallel structure in low-dimensional space; (2) these difference vectors are strongly correlated with task similarity in high-dimensional space; (3) through Fourier-based analysis, these difference activations can be decomposed into task-agnostic, information-enhanced features (low-frequency) and task-specific features (high-frequency).

• We propose **FAST**, a novel cross-task transfer framework that leverages Fourier-based activation steering to transfer knowledge from high-resource to low-resource tasks. We first select a representative and diverse subset from high-resource tasks and apply Fourier-based activation steering to enable effective cross-task transfer without requiring parameter updates or input expansion.

• We conduct extensive experiments in both cross-domain and cross-lingual transfer scenarios. The experimental results demonstrate that our approach consistently outperforms competitive baselines while maintaining high scalability and computational efficiency.

## 2 EMPIRICAL STUDY

The Hopfieldian view of cognition (Hopfield, 1982) posits that neural computation arises from dynamic transformations in population-level neural activity in response to external stimuli. This perspective has inspired mechanistic interpretations of artificial neural networks, where activation steering (Zou et al., 2023) has emerged as a powerful method for analyzing internal model states. By treating intermediate activations as fundamental computation units, activation steering helps reveal high-level concepts and functions encoded within models. In this section, we analyze zero-shot and few-shot prompts across various source and target tasks from the perspective of activations.

### 2.1 EXPERIMENTAL SETUP

Following Liu et al. (2024a), we extract activations from the last token position for both zero-shot and few-shot prompts using Llama3.1-8B (Dubey et al., 2024), as this position aggregates the full semantics of the input sentence. We use AGnews (Zhang et al., 2015), ARC-Easy (Clark et al., 2018),

and SST2 (Socher et al., 2013) as source tasks, and ARC-Challenge (Clark et al., 2018) and Financial Phrasebank (Malo et al., 2014) as target tasks. For each task, we randomly select 500 samples and construct zero-shot and few-shot prompts, with the latter including three randomly chosen examples.

## 2.2 INFORMATION-ENHANCED FEATURES INDUCED BY IN-CONTEXT EXAMPLES EXHIBIT CONSISTENT PATTERNS ACROSS TASKS

Given that few-shot prompts provide richer contextual information and elicit more diverse features, we investigate whether information induced by in-context examples from high-resource tasks can facilitate cross-task transfer via latent space manipulation. To this end, we analyze the activation distributions from both zero-shot and few-shot prompts across various source and target tasks by projecting these high-dimensional activations

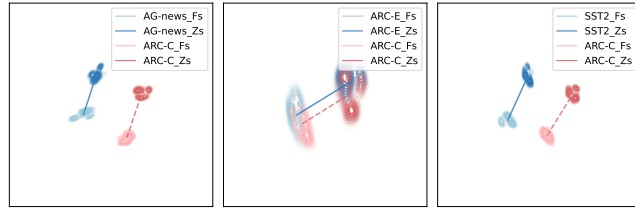

Figure 1: T-SNE visualization of model activations under zero-shot and few-shot prompts. Red and Blue points denote source and target tasks, respectively.

into 2D space using t-SNE (van der Maaten & Hinton, 2008). The results are presented in Figure 1 (More visualization is provided in Figure 12). As we can see, activations from two prompt types form clearly separated clusters within each task. This indicates that the model develops fundamentally distinct internal representations under each prompting strategy. Moreover, the vectors connecting the cluster centers of few-shot and zero-shot activations are nearly parallel across different tasks in the low-dimensional space. This suggests that information-enhanced features follow a consistent direction across tasks, supporting the feasibility of cross-task transfer through latent space steering.

## 2.3 DIFFERENCE ACTIVATION DIRECTIONS BETWEEN FEW-SHOT AND ZERO-SHOT PROMPTS ARE HIGHLY CORRELATED WITH TASK SIMILARITY

Given the consistent directions of information-enhanced features, we examine whether such activations can facilitate cross-task transfer in high-dimensional space. We compute the similarity of the difference activation directions between source and target tasks. For a given layer $l$, we define the difference vector $dv^s(l)$ and $dv^t(l)$ for a source task $s$ and a target task $t$ as:

$$dv^s(l) = \{f^s(i,l) - z^s(i,l)\}_{i=1}^n \quad (1)$$

$$dv^t(l) = \{f^t(i,l) - z^t(i,l)\}_{i=1}^n \quad (2)$$

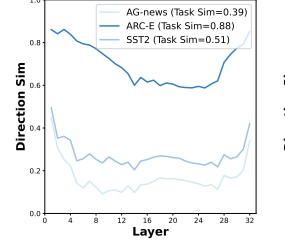 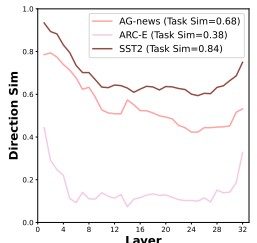

(a) ARC-Challenge    (b) Financial Phrasebank

Figure 2: Similarity of difference activation directions across various task pairs.

where $f(i,l)$ and $z(i,l)$ denote few-shot and zero-shot activations at layer $l$ for each sample $i$, respectively. We then quantify the task similarity $Ts(s,t)$ between source task $s$ and target task $t$ by averaging the pairwise cosine similarities of last-token representations across all layers and samples:

$$Ts(s,t) = \frac{1}{Ln^2} \sum_{l=1}^{L} \sum_{i=1}^{n} \sum_{j=1}^{n} \frac{z_l^s(i) \cdot z_l^t(j)}{|z_l^s(i)||z_l^t(j)|}. \quad (3)$$

As shown in Figure 2, we find a strong positive correlation between task similarity and directions of difference activations, indicating that these vectors retain substantial task-specific information. This makes direct transfer between dissimilar tasks challenging due to the divergence in the activation spaces. Additionally, we find that directional similarity varies most prominently in middle layers, possibly because LLMs encode more abstract features (Skean et al., 2025) in these layers.

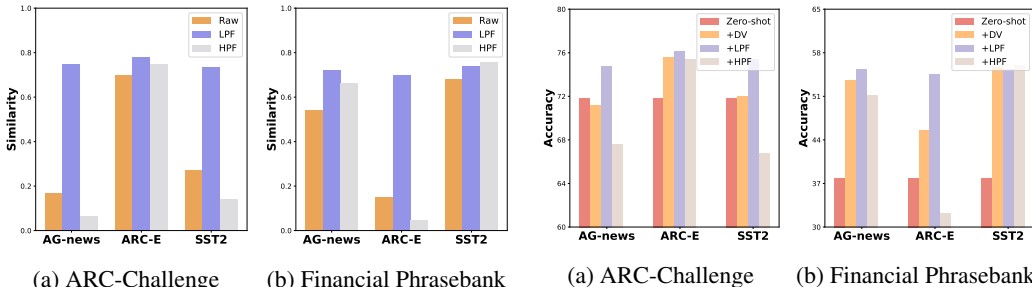

(a) ARC-Challenge   (b) Financial Phrasebank   (a) ARC-Challenge   (b) Financial Phrasebank

Figure 3: Similarity of low-pass and high-pass filtered difference vectors across task pairs.

Figure 4: Accuracy results of injecting difference, low-pass and high-pass filtered activations.

### 2.4 Low-frequency Components Encode Information-enhanced Features and High-frequency Components Retain Task-specific Information

Considering the challenges of directly injecting the difference vectors, we investigate their spectral characteristics across all layers. Inspired by Fourier analysis, where low-pass filters preserve broad trends and high-pass filters capture localized details, we decompose each difference vector $dv$ into low-frequency $dv_{\text{low}}$ and high-frequency components $dv_{\text{high}}$:

$$dv_{\text{low}} = \text{Re}\left[\text{IFFT}\left(\mathbf{M_k} \odot \text{FFT}(dv)\right)\right], \quad dv_{\text{high}} = \text{Re}\left[\text{IFFT}\left((1 - \mathbf{M_k}) \odot \text{FFT}(dv)\right)\right]. \quad (4)$$

Here, $\text{FFT}(\cdot)$ and $\text{IFFT}(\cdot)$ denote the Fast Fourier Transform and its inverse, $\odot$ is the Hadamard product, $\text{Re}[\cdot]$ extracts the real part to eliminate residual imaginary components. The low-pass mask $\mathbf{M}_k \in \{0,1\}^d$, with $d$ being the dimensionality of $dv$, preserves the first $k$ frequency components:

$$\mathbf{M}_k[i] = \begin{cases} 1, & \text{if } i \leq \frac{k}{2} \text{ or } i > d - \frac{k}{2}, \\ 0, & \text{otherwise.} \end{cases} \quad (5)$$

As illustrated in Figure 3, low-pass filtered vectors show high similarity across task pairs, suggesting that low-frequency components capture task-agnostic features. In contrast, high-pass filtered activations are similar only between highly similar tasks, indicating that they encode task-specific information.

We further evaluate the effect of injecting the full difference vector, low-pass filtered components, and high-pass filtered components into LLMs across all layers. The average results across all layers are shown in Figure 4. Our findings reveal that directly injecting the full difference vector strongly correlates with task similarity: it improves performance when tasks are similar, but yields limited or even negative gains when tasks are dissimilar. Moreover, we observe that adding low-frequency activations (*i.e.,* information-enhanced features) consistently improves accuracy across tasks, while injecting high-frequency activations (*i.e.,* task-specific information) only helps when tasks are similar and harms performance when they are dissimilar. This indicates that low-frequency components encode information-enhanced features and high-frequency components retain task-specific information.

## 3 Method

In this section, we introduce **FAST**, a novel framework for cross-task transfer via Fourier-based activation steering in LLMs. We first formalize the problem of cross-task transfer learning, then select influential and diverse samples from high-resource tasks, and finally leverage Fourier-based activations of these samples to adapt LLMs to low-resource tasks through activation steering. The overall framework is illustrated in Figure 5.

### 3.1 Problem Formulation

Cross-task transfer learning aims to enhance the performance of LLMs on low-resource target tasks by leveraging knowledge acquired from high-resource source tasks (Vu et al., 2022; Chatterjee et al., 2024). Let $(x_{s_i}, y_{s_i})_{i=1}^{n} \in D_s$ denote the source task with abundant labeled data, and $(x_{t_i}, y_{t_i})_{i=1}^{n} \in D_t$ represent the target task with limited labeled samples. The process typically

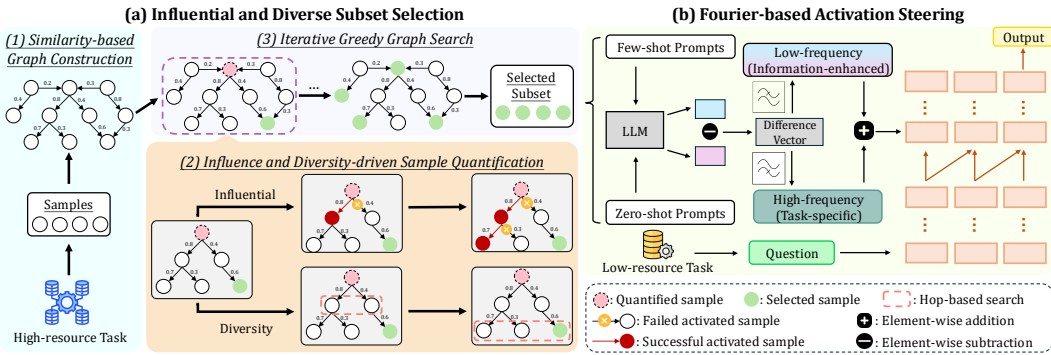

Figure 5: Overview of **FAST**. (a) Influential and diverse subset selection strategy: (1) construct a similarity-based graph from high-resource tasks; (2) quantify each sample's influence and diversity; (3) iteratively select the highest-scoring sample via greedy search to form the final subset. (b) Fourier-based activation steering: decompose the difference vectors between few-shot and zero-shot prompts via Fourier-based filtering into low-frequency (information-enhanced) and high-frequency (task-specific) components, and selectively inject them to steer LLM outputs for low-resource tasks.

involves extracting transferable information $I_s$ from the source task and effectively integrating it into the target task question $x_t$, which can be formalized as:

$$I_s = g(x_{s_1}, y_{s_1}, \cdots, x_{s_n}, y_{s_n}), \quad \hat{y}_t = \text{LLM}(I_s, x_t). \tag{6}$$

Here, $g$ denotes the feature extraction function that captures knowledge from the source tasks, and $\hat{y}_t$ is the prediction of LLMs on the target input.

### 3.2 INFLUENTIAL AND DIVERSE SUBSET SELECTION

Although the source task contains abundant labeled data, utilizing all samples for cross-task transfer is computationally inefficient and unnecessary. To address this, we propose an influential and diverse subset selection strategy to identify representative and diverse samples from the source task. Specifically, we first build a sample graph based on pairwise similarities to model relationships among examples, then measure the influence and diversity score of each sample, and finally employ an iterative greedy search algorithm to select the highest-scoring samples for the final subset.

#### 3.2.1 SIMILARITY-BASED GRAPH CONSTRUCTION

Our subset selection strategy begins by modeling sample relationships using a directed graph. We first encode each sample into a vector using the BGE model (Chen et al., 2024a). These embeddings are used to construct a task-specific directed graph $\mathcal{G} = (\mathbf{V}, \mathbf{E}, \mathbf{P})$, where each vertex $v_i \in \mathbf{V}$ denotes a sample, a directed edge $e(i, j) \in \mathcal{E}$ connects node $v_i$ to $v_j$, and edge weight $p(i, j) \in \mathbf{P}$ is the cosine similarity between the embeddings of the corresponding samples. To reduce structural redundancy, we follow Su et al. (2023) and connect each node to its 150 most similar neighbors.

#### 3.2.2 INFLUENCE AND DIVERSITY-DRIVEN SAMPLE QUANTIFICATION

After constructing the sample graph, we evaluate each sample from two perspectives: (1) its **influence** in activating other samples within the task, and (2) its contribution to the overall **diversity** of the previously selected samples. The pseudo-code is provided in Algorithm 1.

The influence score measures how a sample can propagate information across the graph, which is computed by simulating an information diffusion process. Specifically, we initialize the process by adding the candidate node $v$ into an active set $S_{\text{active}}$. At each step, we randomly select an active node $u \in S_{\text{active}}$ and attempt to activate each of its 1-hop neighbors $w \in N_1(u)$ with success probability $p(u, w)$. Newly activated nodes are added to $S_{\text{active}}$. This iterative process continues until no further propagations occur. The influence score $I(v)$ of node $v$ is defined as the total number of nodes activated during the entire diffusion process. In this way, samples that trigger extensive activation receive higher scores. To ensure the robustness of our method, we follow Zhang et al. (2024) and repeat the simulation 10 times, reporting the average influence score.

The diversity penalty measures the redundancy a candidate node introduces relative to the already selected subset. We perform a hop-based search to examine the $i$-hop neighbor $N_i(v)$ of node $v$ and compute its overlap with $S_{\text{selected}}$. The diversity penalty $D(v)$ of node $v$ is formulated as:

$$D(v) = -\sum_{i=1}^{k} \beta^i \cdot |N_i(v) \cap S_{\text{selected}}|. \tag{7}$$

Here, $\beta$ is a hop-based decay factor that reduces the penalty for overlaps at larger graph distances. Consequently, nodes with minimal overlap have smaller penalties. Finally, we balance the influence score $I(v)$ and diversity penalty $D(v)$ through a hyperparameter $\gamma$ to obtain the overall score $F_{\mathcal{G}}(v)$.

$$F_{\mathcal{G}}(v) = I(v) + \gamma \cdot D(v). \tag{8}$$

### 3.2.3 ITERATIVE GREEDY GRAPH SEARCH

To construct a subset that balances both task representativeness and sample diversity, we use an iterative greedy graph search strategy. The pseudo-code is provided in Algorithm 2. Our approach operates over the sample graph $\mathcal{G}$ and iteratively selects the candidate sample with the highest influence-diversity score. In more detail, the process starts with an empty set. At each iteration, we evaluate all unselected samples using the function $F_{\mathcal{G}}$, and the highest-scoring sample is added to the subset. This procedure is repeated until the subset reaches the desired size.

### 3.3 FOURIER-BASED ACTIVATION STEERING

Inspired by the observation from our empirical study, we propose a Fourier-based activation steering method that transfers high-resource information to low-resource tasks. Our method primarily consists of two components: *activation extraction* and *activation control*.

**Activation Extraction.** The component aims to identify high-level concepts or functional behaviors encoded in LLMs. Specifically, for each sample $i \in D_s$ from the high-resource task, we construct two types of prompts: a zero-shot prompt $z_i$ that contains only the sample, and a few-shot prompt $f_i$ that includes three randomly selected in-context examples. To eliminate instance-specific noise and capture general task-level features, we compute the mean difference activation across all samples.

$$dv^s(l) = \frac{1}{n} \sum_{i=1}^{n} (f^s(i, l) - z^s(i, l)), \tag{9}$$

where $dv^s(l)$ represents the difference vector from the last token's hidden state at layer $l$, and $n$ is the number of samples.

**Activation Control.** This component aims to steer model behaviors by leveraging extracted activations. Motivated by our earlier finding that low-frequency components encode transferable, information-enhanced features and high-frequency components capture task-specific details, we apply Fourier-based filtering to decompose $dv^s$ into $dv^s_{\text{low}}$ and $dv^s_{\text{high}}$ using Equation 4. We then inject the low-frequency component along with the task-similarity weighted high-frequency component into the hidden state at the final token position of a specific layer, which effectively steers model predictions without perturbing previously encoded context. The modified hidden state is computed as:

$$\hat{h}_l = h_l + \lambda \left( dv^s_{\text{low}} + \mathbb{I}[Ts(s,t) > \epsilon] \cdot Ts(s,t) \cdot dv^s_{\text{high}} \right), \tag{10}$$

where $\mathbb{I}$ is the indicator function, $\lambda$ is the injection strength, $h_l$ is the hidden state at layer $l$, $Ts(s,t)$ represents the task similarity between source $s$ and target task $t$, and $\epsilon$ is the similarity threshold. This approach enhances cross-task transfer by emphasizing task-agnostic and information-enhanced features and adaptively incorporating task-specific activations for sufficiently similar task pairs.

## 4 EXPERIMENTS

### 4.1 EXPERIMENTAL SETUP

**Datasets.** In this paper, we evaluate our method in both cross-domain and cross-lingual transfer settings. For cross-domain experiments, we follow Chatterjee et al. (2024) and use seven source

Table 1: Performance comparisons in the cross-domain transfer scenarios.

| Model | Method | | ARC-C | FPB | MedMCQA | SciQ | Social-i-QA | Average |
|---|---|---|---|---|---|---|---|---|
| **Llama 3.1-8B** | **Prompting** | Zero-shot | 71.80 | 37.80 | 49.40 | 84.40 | 55.40 | 59.76 |
| | | Few-shot Random | $69.31_{\pm2.66}$ | $48.51_{\pm3.65}$ | $48.69_{\pm2.23}$ | $84.63_{\pm2.85}$ | $59.14_{\pm1.71}$ | $62.06_{\pm2.62}$ |
| | | Few-shot TopK | $69.54_{\pm2.42}$ | $48.63_{\pm6.07}$ | $48.69_{\pm1.77}$ | $84.86_{\pm2.28}$ | $59.37_{\pm2.35}$ | $62.22_{\pm2.98}$ |
| | | Few-shot DPP | $69.74_{\pm2.85}$ | $48.89_{\pm5.96}$ | $49.74_{\pm2.12}$ | $85.26_{\pm2.26}$ | $59.60_{\pm1.36}$ | $62.65_{\pm2.91}$ |
| | **PEFT** | QLoRA | $69.31_{\pm3.60}$ | $52.33_{\pm6.04}$ | $50.06_{\pm3.24}$ | $83.97_{\pm4.53}$ | $57.77_{\pm4.16}$ | $62.69_{\pm4.31}$ |
| | | AdaLoRA | $69.74_{\pm3.55}$ | $52.69_{\pm6.76}$ | $50.23_{\pm3.26}$ | $84.21_{\pm4.19}$ | $57.89_{\pm4.64}$ | $62.95_{\pm4.48}$ |
| | **Activation Steering** | ICV | $72.63_{\pm0.95}$ | $54.51_{\pm4.40}$ | $52.77_{\pm1.11}$ | $87.46_{\pm2.45}$ | $61.54_{\pm1.62}$ | $65.78_{\pm2.11}$ |
| | | SEA | $73.49_{\pm0.55}$ | $55.29_{\pm4.49}$ | $53.29_{\pm1.04}$ | $88.00_{\pm2.14}$ | $62.11_{\pm1.28}$ | $66.44_{\pm1.90}$ |
| | | **FAST** | $\mathbf{76.14}_{\pm0.52}$ | $\mathbf{59.26}_{\pm3.90}$ | $\mathbf{56.09}_{\pm0.67}$ | $\mathbf{90.43}_{\pm1.03}$ | $\mathbf{64.57}_{\pm1.18}$ | $\mathbf{69.30}_{\pm1.46}$ |
| **Qwen 2.5-7B** | **Prompting** | Zero-shot | 82.80 | 85.20 | 52.00 | 89.60 | 76.00 | 77.12 |
| | | Few-shot Random | $85.80_{\pm1.32}$ | $86.89_{\pm1.66}$ | $54.37_{\pm1.20}$ | $89.63_{\pm1.55}$ | $77.00_{\pm1.26}$ | $78.74_{\pm1.40}$ |
| | | Few-shot TopK | $86.31_{\pm0.80}$ | $87.69_{\pm1.99}$ | $54.54_{\pm0.71}$ | $90.03_{\pm1.35}$ | $76.97_{\pm0.88}$ | $79.11_{\pm1.15}$ |
| | | Few-shot DPP | $86.31_{\pm0.67}$ | $87.74_{\pm1.46}$ | $54.60_{\pm0.76}$ | $89.94_{\pm1.66}$ | $76.83_{\pm1.16}$ | $79.08_{\pm1.14}$ |
| | **PEFT** | QLoRA | $86.34_{\pm3.34}$ | $87.06_{\pm4.03}$ | $55.51_{\pm3.72}$ | $88.71_{\pm4.54}$ | $77.31_{\pm3.80}$ | $78.99_{\pm3.89}$ |
| | | AdaLoRA | $86.31_{\pm3.67}$ | $87.71_{\pm4.01}$ | $55.86_{\pm3.89}$ | $89.49_{\pm4.23}$ | $77.60_{\pm3.75}$ | $79.39_{\pm3.91}$ |
| | **Activation Steering** | ICV | $88.49_{\pm1.04}$ | $90.43_{\pm1.81}$ | $57.34_{\pm1.40}$ | $91.91_{\pm1.48}$ | $80.11_{\pm1.15}$ | $81.66_{\pm1.38}$ |
| | | SEA | $89.14_{\pm0.99}$ | $90.74_{\pm1.77}$ | $58.11_{\pm1.50}$ | $92.66_{\pm1.20}$ | $80.46_{\pm1.38}$ | $82.22_{\pm1.37}$ |
| | | **FAST** | $\mathbf{92.20}_{\pm1.01}$ | $\mathbf{93.80}_{\pm0.68}$ | $\mathbf{60.86}_{\pm1.51}$ | $\mathbf{95.23}_{\pm0.66}$ | $\mathbf{83.46}_{\pm0.86}$ | $\mathbf{85.11}_{\pm0.94}$ |

Table 2: Performance comparisons in the cross-lingual transfer scenarios.

| Method | | de | en | es | fr | ja | zh | Average |
|---|---|---|---|---|---|---|---|---|
| **Prompting** | Zero-shot | 84.40 | 66.00 | 81.60 | 86.60 | 38.60 | 30.80 | 64.67 |
| | Few-shot Random | $85.64_{\pm5.30}$ | $59.36_{\pm15.58}$ | $83.48_{\pm9.58}$ | $81.40_{\pm9.65}$ | $39.28_{\pm7.11}$ | $37.36_{\pm6.07}$ | $64.42_{\pm8.88}$ |
| | Few-shot TopK | $86.12_{\pm5.31}$ | $61.56_{\pm15.65}$ | $83.68_{\pm9.73}$ | $81.75_{\pm9.99}$ | $39.40_{\pm7.86}$ | $38.36_{\pm5.58}$ | $65.14_{\pm9.02}$ |
| | Few-shot DPP | $86.36_{\pm5.07}$ | $64.48_{\pm15.13}$ | $85.40_{\pm8.42}$ | $83.90_{\pm8.80}$ | $39.84_{\pm8.59}$ | $38.72_{\pm6.81}$ | $66.45_{\pm8.80}$ |
| **PEFT** | QLoRA | $85.16_{\pm6.25}$ | $63.48_{\pm20.01}$ | $82.16_{\pm13.17}$ | $79.35_{\pm13.32}$ | $37.72_{\pm13.05}$ | $35.92_{\pm8.76}$ | $63.97_{\pm12.43}$ |
| | AdaLoRA | $85.64_{\pm6.07}$ | $65.08_{\pm19.85}$ | $83.00_{\pm14.07}$ | $80.20_{\pm14.43}$ | $38.80_{\pm11.92}$ | $36.80_{\pm8.94}$ | $64.92_{\pm12.55}$ |
| **Activation Steering** | ICV | $90.16_{\pm2.57}$ | $82.80_{\pm3.92}$ | $91.96_{\pm2.06}$ | $91.70_{\pm2.23}$ | $44.32_{\pm4.11}$ | $41.16_{\pm5.55}$ | $73.68_{\pm3.41}$ |
| | SEA | $90.44_{\pm2.63}$ | $84.16_{\pm2.97}$ | $92.20_{\pm1.62}$ | $91.85_{\pm1.63}$ | $45.08_{\pm3.70}$ | $42.24_{\pm5.55}$ | $74.33_{\pm3.02}$ |
| | **FAST** | $\mathbf{92.48}_{\pm2.08}$ | $\mathbf{89.04}_{\pm2.11}$ | $\mathbf{95.20}_{\pm0.40}$ | $\mathbf{95.25}_{\pm0.43}$ | $\mathbf{48.12}_{\pm2.61}$ | $\mathbf{44.88}_{\pm4.17}$ | $\mathbf{77.50}_{\pm1.97}$ |

domains and five target domains. The source domains include: ARC-Easy (Clark et al., 2018), AG-news (Zhang et al., 2015), BoolQ (Clark et al., 2019), Commonsense-QA (Talmor et al., 2019), MNLI (Williams et al., 2018), QQP (Sharma et al., 2019), and SST2 (Socher et al., 2013). Following previous work (Chatterjee et al., 2024), we select ARC-Challenge (Clark et al., 2018), Financial-Phrasebank (Malo et al., 2014), MedMCQA (Pal et al., 2022), SciQ (Auer et al., 2023), and Social-i-QA (Sap et al., 2019) as target domains. For cross-lingual settings, we conduct experiments on the MARC (Keung et al., 2020) dataset, which covers six languages. Due to computational constraints, we follow previous work (Chatterjee et al., 2024) and randomly sample 500 examples from each target domain as the test set. Detailed descriptions of the datasets are provided in Appendix E.

**Baselines.** We select several representative approaches for comparison, including prompting methods (*i.e.,* Zero-shot, Few-shot Random, Few-shot TopK (Liu et al., 2022b), Few-shot DPP (Ye et al., 2023)), parameter-efficient fine-tuning methods (*i.e.,* QLoRA (Dettmers et al., 2023) and AdaLoRA (Zhang et al., 2023)), and activation steering methods (*i.e.,* ICV (Liu et al., 2024a) and SEA (Qiu et al., 2024)). Detailed descriptions of these baselines are presented in Appendix F.

**Implementation Details.** Our experiments are conducted using Llama3.1-8B (Dubey et al., 2024) and Qwen2.5-7B (Yang et al., 2025). In our subset selection strategy, we set the subset size $n$ to 20, the hop-based decay factor $\alpha$ to 0.2, and the balanced parameter $\gamma$ between influence and diversity to 0.5. For activation steering, we inject the activations into the final token's hidden state. The injection layer is determined based on performance on the validation set, the injection strength $\lambda$ is set to 0.2, the similarity threshold $\epsilon$ is set to 0.6, and the preserved frequency component $k$ is set to $d/2$. We use accuracy as the evaluation metric. All experiments are computed on 8 A800 GPUs.

## 4.2 EXPERIMENTAL RESULTS

**Cross-domain transfer scenarios.** Table 1 shows the results for the cross-domain transfer setting. Detailed results are provided in Table 8 and Table 9. We observe that the performance of the

cross-task few-shot prompting is highly dependent on the similarity between source and target domains. When domains are closely related (*e.g.,* ARC-Easy → ARC-Challenge), incorporating source-domain examples improves performance. In contrast, for dissimilar domain pairs (*e.g.,* ARC-Easy → MedMCQA), such examples introduce noise and lead to performance degradation. For parameter-efficient fine-tuning (PEFT) methods, performance varies more substantially across different task pairs. Fine-tuning with source task examples generalizes effectively to similar target tasks, but impairs performance on dissimilar ones. All activation steering methods facilitate effective cross-domain transfer due to the consistent directions of information-enhanced features in the latent space across different domains. Among these, **FAST** consistently outperforms all baselines across all domain pairs. This is because our approach applies the Fourier transformation method to disentangle information-enhanced and domain-specific activations within the difference vectors. By injecting both task-agnostic enhanced features and similarity-weighted domain-specific information, **FAST** enhances general model capability while avoiding the injection of irrelevant domain noise.

**Cross-lingual transfer scenarios.** Table 2 presents the results for the cross-lingual transfer setting. Detailed statistics are provided in Table 10. We find that the performance of cross-lingual few-shot prompting is strongly influenced by the linguistic similarity between source and target languages. For closely related language pairs (*e.g.,* French → German), incorporating cross-lingual examples generally improves performance. In contrast, for distant language pairs (*e.g.,* English → Chinese), such demonstrations often introduce noise and lead to negative transfer. Similar to the cross-domain scenario, PEFT methods amplify this phenomenon, exhibiting higher variance across language pairs. Furthermore, activation steering methods effectively facilitate cross-lingual transfer by operating in the latent space, achieving robust performance across diverse language pairs. Among these, **FAST** achieves the best performance by leveraging the Fourier transformation to decouple information-enhanced features via low-pass filtering. Furthermore, it selectively injects high-frequency domain-specific features when source and target tasks exhibit high similarity, which enables more effective and stable latent space steering for cross-lingual transfer.

## 4.3 ABLATION STUDY

Our approach introduces two key components: (1) Influential and diverse subset selection, and (2) Fourier-based activation steering. To verify the effectiveness of each component, we conduct ablation studies on three target tasks: ARC-Challenge, Financial Phrasebank, and MedM-CQA. We also compare our subset selection method with two established approaches (*i.e.,* Vote-k (Su et al., 2023) and IDEAL (Zhang et al., 2024)). The results are presented in Table 3. We observe that removing any component leads to performance degradation across all tasks, con-

Table 3: Ablation study.

| Dataset | ARC-C | FPB | MedMCQA |
|---|---|---|---|
| **FAST** | **76.14** | **59.26** | **56.09** |
| *Subset Selection* | | | |
| w/o Influence Score | 75.62 | 56.91 | 54.83 |
| w/o Diversity Penalty | 74.91 | 57.38 | 55.29 |
| w/o Both | 73.72 | 54.29 | 53.18 |
| Vote-k | 75.46 | 57.89 | 55.35 |
| IDEAL | 75.69 | 58.04 | 55.04 |
| *Activation Steering* | | | |
| w/o Information-enhanced Activation | 72.25 | 48.32 | 51.89 |
| w/o Task-specific Activation | 75.48 | 56.59 | 57.71 |
| w/o Both | 71.80 | 37.80 | 49.40 |

firming that both elements are essential to our method. Notably, the removal of information-enhanced activation causes the most significant performance drop, indicating that these activations provide the most important signals for cross-task transfer. Furthermore, our selection strategy outperforms both Vote-k and IDEAL, demonstrating the effectiveness of the proposed approach.

## 4.4 THE EFFICIENCY OF FAST

In this part, we analyze the computational efficiency of our method in comparison to baseline methods. As shown in Table 4, our approach demonstrates significant advantages in both time complexity and runtime. **FAST** maintains the same time complexity, $O(n^2)$, as zero-shot prompting, which is substantially more efficient than few-shot methods that exhibit $O((d+n)^2)$ complexity due to their longer in-

Table 4: Efficiency comparison of different methods. "T.C." denotes time complexity, where $n$ and $d$ represent the length of the question and demonstrations, respectively.

| Method | Inference T.C. | Preprocess Time (s) | Training Time (s) | Inference Time (s) | Total Time (s) |
|---|---|---|---|---|---|
| Zero-shot | $\mathcal{O}(n^2)$ | 0 | 0 | 212 | 212 |
| Few-shot Random | $\mathcal{O}((d+n)^2)$ | 0 | 0 | 451 | 451 |
| Few-shot DPP | $\mathcal{O}((d+n)^2)$ | 138 | 0 | 454 | 592 |
| AdaLoRA | $\mathcal{O}(n^2)$ | 0 | 332 | 215 | 547 |
| **FAST** | $\mathcal{O}(n^2)$ | 172 | 0 | 221 | 393 |

Table 5: Experiments on generation tasks.

| Method | | XSum | GSM8K | LiveCodebench | GPQA | Average |
|---|---|---|---|---|---|---|
| **Prompting** | Zero-shot | 28.32 | 76.65 | 16.67 | 35.29 | 39.73 |
| | Few-shot Random | $27.46_{\pm3.48}$ | $79.91_{\pm4.95}$ | $11.74_{\pm2.94}$ | $31.14_{\pm5.02}$ | $37.56_{\pm4.10}$ |
| | Few-shot TopK | $28.04_{\pm2.76}$ | $80.89_{\pm3.47}$ | $13.31_{\pm2.34}$ | $31.68_{\pm4.89}$ | $38.48_{\pm3.37}$ |
| | Few-shot DPP | $29.05_{\pm3.18}$ | $81.20_{\pm2.95}$ | $13.89_{\pm2.49}$ | $32.97_{\pm4.07}$ | $39.28_{\pm3.17}$ |
| **PEFT** | QLoRA | $23.42_{\pm6.97}$ | $69.26_{\pm5.08}$ | $12.79_{\pm3.29}$ | $29.97_{\pm6.38}$ | $33.86_{\pm5.43}$ |
| | AdaLoRA | $24.01_{\pm6.14}$ | $70.91_{\pm5.26}$ | $11.92_{\pm4.05}$ | $30.48_{\pm7.29}$ | $34.33_{\pm5.68}$ |
| **Activation Steering** | ICV | $31.82_{\pm2.48}$ | $83.93_{\pm2.18}$ | $18.20_{\pm2.11}$ | $37.25_{\pm3.48}$ | $42.80_{\pm2.56}$ |
| | SEA | $32.49_{\pm1.97}$ | $84.91_{\pm2.11}$ | $18.78_{\pm1.94}$ | $38.82_{\pm3.05}$ | $43.75_{\pm2.27}$ |
| | **FAST** | $\mathbf{34.41}_{\pm1.53}$ | $\mathbf{86.13}_{\pm1.74}$ | $\mathbf{20.55}_{\pm2.35}$ | $\mathbf{40.48}_{\pm3.75}$ | $\mathbf{45.39}_{\pm2.34}$ |

put sequences. This is because our method injects activations into the model's latent space during the forward pass without adding additional tokens to the input. In terms of actual runtime, **FAST** requires only 393 seconds in total, which includes 172 seconds for preprocessing (subset selection and activation extraction) and 221 seconds for inference. This represents a notable improvement over few-shot methods and PEFT approaches, demonstrating that **FAST** achieves effective cross-task transfer while maintaining computational efficiency.

## 4.5 DETAILED ANALYSIS

In this section, we present a detailed analysis of the proposed method. Unless otherwise stated, we conduct experiments using ARC-Challenge as the target task.

**FAST performs well on different-scale LLMs.** We conduct experiments on Qwen-series LLMs ranging from 0.5B to 32B in Figure 6, with additional results in Figure 9. Notably, activation steering methods consistently outperform few-shot prompting across all model sizes. Among these, **FAST** achieves the best performance by explicitly disentangling information-enhanced features from task-specific activations, enabling more effective and robust cross-task transfer.

**FAST demonstrates strong scalability.** To evaluate the scalability of our proposed method, we conduct experiments using different numbers of examples from the source task, as shown in Figure 7 (additional results in Figure 10). We find that the performance of cross-task few-shot learning initially improves with more examples, but eventually plateaus or even drops when too many examples are used. This can be attributed to the limited long-context capability of LLMs, which hinders the effective use of large-scale high-resource data. In contrast, activation steering methods show a positive correlation between the number of examples and model performance. With more demonstrations, these methods better isolate instance-level variations and extract more general task-level features, leading to more effective cross-task transfer. In particular, **FAST** demonstrates the strongest scalability among all activation steering methods due to its decoupled activation injection method.

**Optimal performance of FAST at middle layers with moderate injection strength.** As shown in Figure 8 (additional results in Figure 11), both the injection layer and the injection strength significantly affect the performance of **FAST**. We find that injecting activations at middle layers yields the best results, indicating that these layers encode richer features that are beneficial for cross-task transfer. Furthermore, **FAST** achieves optimal results when the injection strength is set to 0.2. Higher values tend to disrupt the model's inherent representations, while lower values produce insufficient signals to effectively guide the model's behavior toward the target task.

**Our method performs well on generation tasks.** To evaluate the generalizability of our proposed method beyond classification tasks, we conduct experiments on a variety of generation tasks on Qwen2.5-7B. We consider four generation benchmarks: XSum (Narayan et al., 2018) for summarization, GSM8K (Cobbe et al., 2021) for mathematical reasoning, GPQA (Rein et al., 2024) for scientific question answering, and LiveCodeBench (Jain et al., 2025) for code generation. The results are presented in Table 5. We inject activations into the hidden state of the first token, as it steers the subsequent generation process while minimizing disruption to the model's inherent generation behavior.

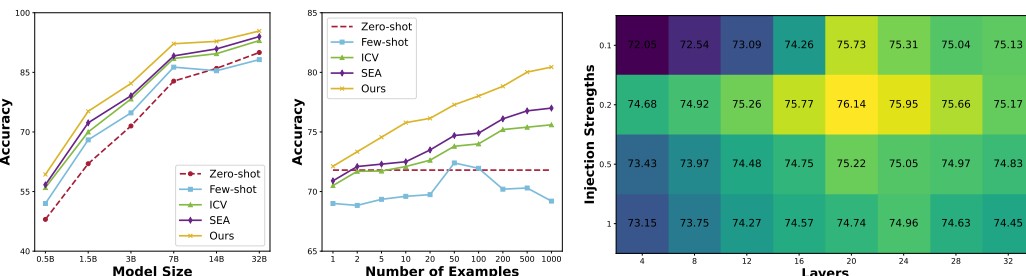

Figure 6: Model sizes    Figure 7: Scalability    Figure 8: Injection layers and strengths

We observe that incorporating cross-task examples via cross-domain in-context learning often leads to performance degradation in these generation tasks. This suggests that, unlike in classification tasks, in-context examples from dissimilar tasks may introduce noise that disrupts the generation process. Besides, parameter-efficient fine-tuning methods also struggle to generalize effectively, likely due to their limited capacity to adapt to the generation tasks from dissimilarity tasks. Notably, activation steering methods consistently outperform both prompting-based and parameter-efficient fine-tuning baselines. Among these, our method achieves the best performance across all tasks. The consistent improvements highlight **FAST** can effectively inject transferable information-enhanced activations.

## 5 RELATED WORK

**Transfer Learning.** Transfer learning offers a promising solution to alleviate the scarcity of labeled data in low-resource tasks by leveraging knowledge from high-resource tasks. Existing transfer learning approaches for LLMs can be broadly categorized into two types: continuous and discrete cross-task transfer. Continuous methods (Vu et al., 2022; Li et al., 2022; Lv et al., 2024) learn shared continuous soft prompts from source tasks and apply them to the target tasks. While effective, these approaches require fine-tuning and often generalize poorly. On the other hand, discrete methods (Tanwar et al., 2023; Cahyawijaya et al., 2024; Li et al., 2023b; Chatterjee et al., 2024) incorporate high-resource examples into LLM inputs to solve low-resource tasks without parameter updates. However, such an approach suffers from limitations in robustness, scalability, and efficiency. To address these issues, we propose a novel approach to extract activations from high-resource tasks and inject them into low-resource tasks, which eliminates the need for fine-tuning or input expansion.

**Activation Steering.** Activation steering is an established technique that treats internal representations as fundamental units for analysis and manipulation within neural networks. It has been applied across various scenarios, including model alignment (Liu et al., 2024b), personality modeling (Cao et al., 2024), instruction following (Stolfo et al., 2025), hallucination mitigation (Li et al., 2023a; Arditi et al., 2024), safety enhancement (Liu et al., 2024a), and reasoning improvement (Højer et al., 2025; Tang et al., 2025). In this work, we adopt activation steering to transfer knowledge from data-sufficient to data-scarce tasks, providing a new pathway for effective cross-task generalization.

## 6 CONCLUSION

In this work, we explored the potential of achieving cross-task transfer in LLMs via latent space steering. Through empirical analysis, we found consistent activation patterns between few-shot and zero-shot prompts across tasks. Besides, we observed that the difference activations can be decomposed via Fourier transformation into information-enhanced (low-frequency) and task-specific (high-frequency) components. Based on these insights, we proposed **FAST**, a Fourier-based activation steering framework that transfers knowledge from high-resource to low-resource tasks without fine-tuning or context expansion. Extensive experiments in cross-domain and cross-lingual settings demonstrated that our method consistently outperformed existing approaches.

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

---

**Algorithm 1** Influence and Diversity-driven Sample Quantification$(\mathcal{G}, S_{\text{selected}}, N_i(\cdot), v, k, \beta, \gamma)$

---

**Inputs:**
    Sample directed graph $\mathcal{G} = (\mathbf{V}, \mathbf{E}, \mathbf{P})$, $i$-hop neighbibor function $N_i(\cdot)$, Current selected sample subset $S_{\text{selected}}$, Sample node $v$, Neighborhood depth $k$, Hop-based decay factor $\beta$, balance hyper-parameter between diversity and influence $\gamma$.

**Initialize:**
    $D(v) = 0$, $I(v) = 0$, $S_{\text{active}} \leftarrow v$, $S_{\text{visited}} \leftarrow \emptyset$

**while** $S_{\text{active}} \neq \emptyset$ **do**                                             ▷ Influencial Calculation
    Choose a sample node $u \in S_{\text{active}}$
    **for** each neighbor $w \in N_1(u)$ **do**
        Select edge $(u, w)$ with probability $p(u, w)$
        **if** edge $(u, w)$ is selected **and** $w \notin S_{\text{visited}}$ **then**
            $S_{\text{active}} \leftarrow S_{\text{active}} \cup w$, $S_{\text{visited}} \leftarrow S_{\text{visited}} \cup w$
        **end if**
    **end for**
    $S_{\text{active}} \leftarrow S_{\text{active}} \setminus u$
**end while**
$I(v) = |S_{\text{visited}}|$
**for** $i = 1$ to $k$ **do**                                                    ▷ Diversity Calculation
    Search $i$-hop neighbors of sample node $v$: $N_i(v)$;
    Compute overlap between $i$-hop neighbors and $S_{\text{selected}}$: $o_i \leftarrow |N_i(v) \cap S_{\text{selected}}|$
    $D(v) \leftarrow D(v) - \beta^i \cdot o_i$
**end for**
**return** Sample node $v$ evaluation function: $F_{\mathcal{G}}(v) \leftarrow I(v) + \gamma \cdot D(v)$

---

**Algorithm 2** Iterative Greedy Graph Search$(\mathcal{G}, S, n)$

---

**Inputs:**
    Sample directed graph $\mathcal{G} = (\mathbf{V}, \mathbf{E}, \mathbf{P})$, Initial sample subset $S_0$, Selected sample subset size $n$.

**Initialize:**
    $\mathcal{S}_0 \rightarrow \emptyset$, $i = 0$, Sample node evaluation function $f_{\mathcal{G}} : \mathbf{V} \mapsto \mathbb{R}$ based on Algorithm 1

**while** $i < n$ **do**
    $v^* \leftarrow \underset{v \in \mathbf{V} \setminus \mathcal{S}_i}{\arg\max} F_{\mathcal{G}}(v)$
    $\mathcal{S}_{i+1} \leftarrow \mathcal{S}_i \cup v^*$
    $i \leftarrow i + 1$
**end while**
**return** $\mathcal{S}_n$.

---

## A   Usage of LLMs

In this paper, Large Language Models are used solely for polishing the writing.

## B   Reproducibility Statement

Our code is provided in the anonymous link to facilitate reproducibility.

## C   Ethics Statement

This work adheres to the ICLR Code of Ethics. No ethical issues arise from this research.

## D   Additional Experiments

### D.1   Hyperparameter Analysis

**FAST** includes a few hyperparameters to tune. In this section, we present a detailed analysis of their impact on model performance. For the influential and diverse subset selection strategy, we examine the hop-based decay factor $\alpha$ and the trade-off parameter $\gamma$ that balances diversity and influence. The results are illustrated in Figure 13a and Figure 13b. As we can see, setting $\alpha$ too small or too large

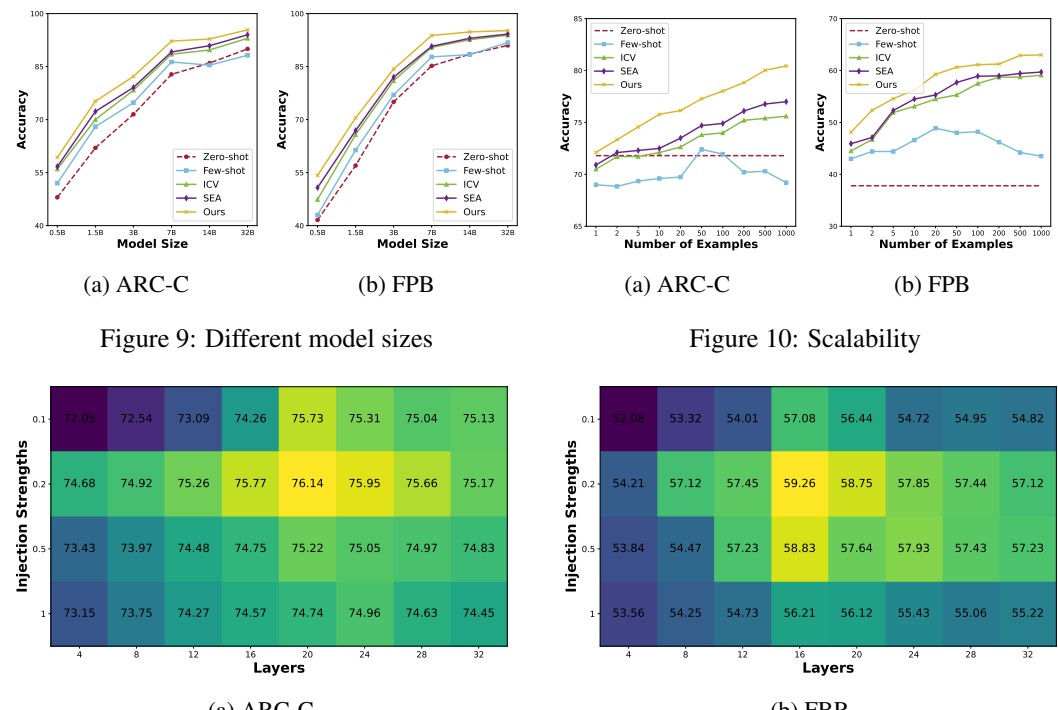

(a) ARC-C       (b) FPB         (a) ARC-C       (b) FPB

Figure 9: Different model sizes           Figure 10: Scalability

(a) ARC-C                  (b) FBR

Figure 11: Different injection layers and strengths

leads to a performance drop. If $\alpha$ is too small, the method may overlook meaningful connections between nodes that are indirectly linked. In contrast, an excessively large $\alpha$ would decrease the positional relationships between nodes within graphs. As for balanced hyperparameter $\gamma$, we find that choosing an appropriate value helps achieve a good balance, resulting in a subset that is both representative and diverse.

For the activation steering component, we investigate the effect of frequency cutoff $k$, similarity threshold $\epsilon$ and injection position, with results presented in Figure 13c, Figure 13d, and Table 6. We find that the frequency cutoff $k = d/2$ yields the best performance, as it effectively separates low- and high-frequency components. Besides, the optimal similarity threshold is 0.6. Performance remains relatively stable across values from 0.4 to 0.7, with our method consistently outperforming all baselines in this range. In addition, our finding reveals that injecting activations at the last token position consistently yields the best performance across models and target tasks. This is because the final token position aggregates sufficient contextual information, and modifications at this position can directly affect the output generation without disrupting the encoding of earlier tokens.

## D.2 EXPERIMENTS ON MULTI-MODAL TASKS.

To further evaluate our proposed method, we conduct multimodal tasks using MathVista (Lu et al., 2024), MMStar (Chen et al., 2024b) and MMMU (Yue et al., 2024) datasets using Qwen2.5-VL-7B-Instruct (Bai et al., 2025). The results are presented in Table 7. The results show that our method consistently achieves the best performance on these multimodal reasoning tasks, demonstrating its broad applicability.

## E  DATASET DETAILS

In this part, we provide detailed descriptions of the datasets used in our experiments, covering both cross-domain and cross-lingual scenarios.

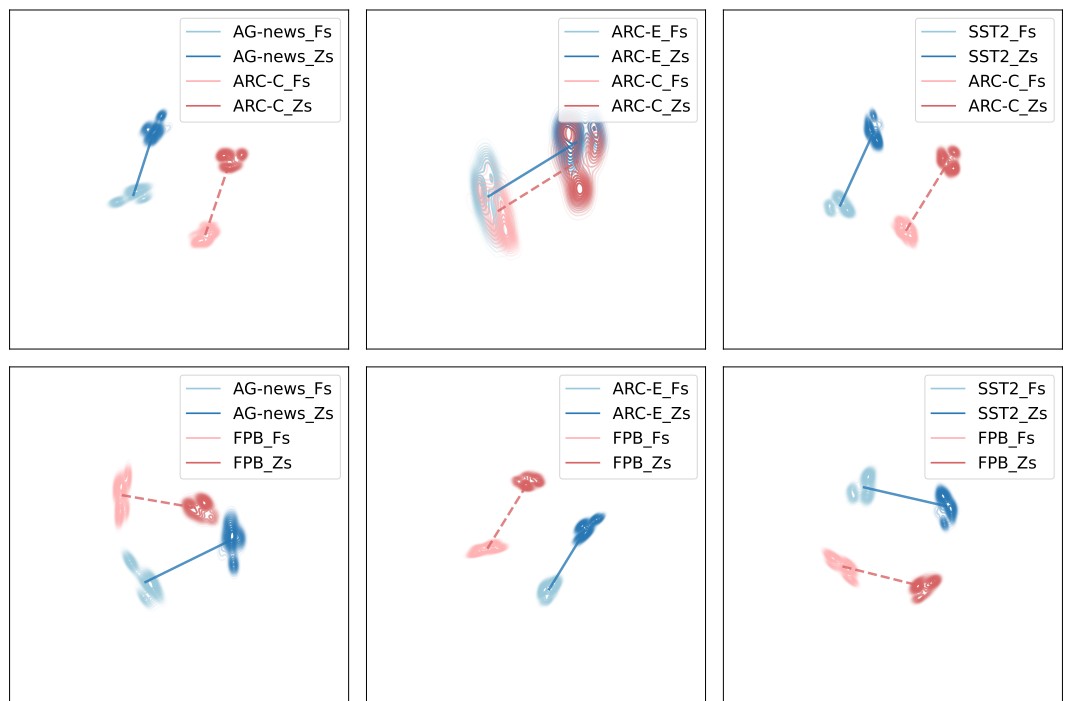

Figure 12: T-SNE projection of zero-shot and few-shot prompts across different source and target tasks.

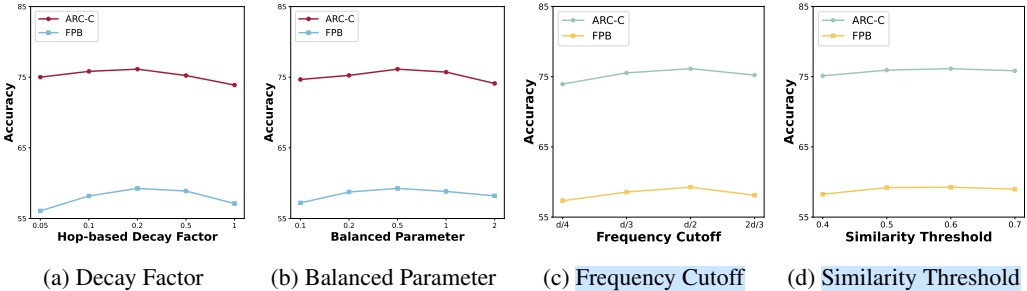

(a) Decay Factor  (b) Balanced Parameter  (c) Frequency Cutoff  (d) Similarity Threshold

Figure 13: Hyperparameter Analysis

### E.1 CROSS-DOMAIN SCENARIOS

• **ARC-Easy**: ARC-Easy (Clark et al., 2018) is a multiple-choice question-answering dataset, which consists of simple science exam questions from grade 3 to grade 9. These questions are designed to be straightforward and require basic knowledge.

• **AG-news**: AG-news (Zhang et al., 2015) is a news topic classification dataset, which is constructed by collecting article titles and descriptions from the four main categories: World, Sports, Business, and Sci/Tech.

• **BoolQ**: BoolQ (Clark et al., 2019) is a reading comprehension dataset with yes/no questions. The task requires answering these binary questions based on the given passages.

• **Commonsense-QA**: Commonsense-QA (Talmor et al., 2019) is a multiple-choice question answering dataset that requires different types of commonsense knowledge to find the correct answers.

• **MNLI**:The Multi-Genre Natural Language Inference (MNLI) (Williams et al., 2018) is a crowd-sourced collection of 433k sentence pairs annotated with textual entailment information. The task is to classify the relationship between two sentences as entailment, contradiction, or neutral.

Table 6: Performance comparison across different injection positions.

| Position | Random | All | First | Last |
|---|---|---|---|---|
| Target Task: ARC-C | | | | |
| Llama3.1-8B | 73.12 | 74.69 | 76.02 | **76.14** |
| Qwen2.5-7B | 89.60 | 89.17 | 91.63 | **92.20** |
| Target Task: FBR | | | | |
| Llama3.1-8B | 55.28 | 55.73 | 58.48 | **59.26** |
| Qwen2.5-7B | 89.75 | 87.68 | 91.86 | **93.80** |

Table 7: Experiments on multi-modal tasks.

| Method | | MathVista | MMStar | MMMU | Average |
|---|---|---|---|---|---|
| **Prompting** | Zero-shot | 68.52 | 64.12 | 58.08 | 63.57 |
| | Few-shot Random | $64.43_{\pm 2.14}$ | $62.12_{\pm 3.28}$ | $59.71_{\pm 2.54}$ | $62.09_{\pm 2.65}$ |
| | Few-shot TopK | $64.48_{\pm 2.39}$ | $62.28_{\pm 2.56}$ | $59.23_{\pm 2.47}$ | $62.00_{\pm 2.47}$ |
| | Few-shot DPP | $64.91_{\pm 2.94}$ | $62.54_{\pm 3.19}$ | $60.18_{\pm 2.39}$ | $62.54_{\pm 2.84}$ |
| **PEFT** | QLoRA | $60.53_{\pm 5.12}$ | $65.23_{\pm 7.19}$ | $58.24_{\pm 4.17}$ | $61.33_{\pm 5.49}$ |
| | AdaLoRA | $62.91_{\pm 6.72}$ | $65.26_{\pm 6.24}$ | $58.46_{\pm 4.28}$ | $62.21_{\pm 5.75}$ |
| **Activation Steering** | ICV | $71.17_{\pm 1.48}$ | $65.19_{\pm 2.23}$ | $61.24_{\pm 1.82}$ | $65.87_{\pm 1.84}$ |
| | SEA | $71.24_{\pm 1.24}$ | $66.50_{\pm 2.45}$ | $61.85_{\pm 1.95}$ | $66.53_{\pm 1.88}$ |
| | **FAST** | $\mathbf{73.34}_{\pm 1.31}$ | $\mathbf{68.83}_{\pm 1.57}$ | $\mathbf{64.12}_{\pm 1.72}$ | $\mathbf{68.76}_{\pm 1.53}$ |

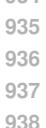

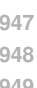

• **QQP**: Quora Question Pairs (QQP) (Sharma et al., 2019) is a natural language understanding dataset comprising over 400k question pairs. Each question pair is annotated with a binary label indicating whether the two questions are duplicates of each other.

• **SST2**: The Stanford Sentiment Treebank (SST2) (Socher et al., 2013) is a binary sentiment classification dataset, which contains the movie reviews labeled as either positive or negative.

• **ARC-Challenge**: ARC-Challenge (Clark et al., 2018) is a more difficult version of ARC-Easy. It also includes science exam questions for grades 3 to 9, but requires deeper reasoning and advanced problem-solving strategies.

• **Financial Phrasebank**: Financial Phrasebank (Malo et al., 2014) is a sentiment analysis dataset focused on financial news, which consists of financial news articles annotated with sentiment labels such as positive, negative, or neutral.

• **MedMCQA**: MedMCQA (Pal et al., 2022) is a large-scale, multiple-choice question answering dataset, designed to address real-world medical entrance exam questions.

• **SciQ**: SciQ (Auer et al., 2023) is a multiple-choice question answering dataset comprising science exam questions in the fields of physics, chemistry, and biology.

• **Social-i-QA**: Social-i-QA (Sap et al., 2019) is a question-answering benchmark designed to evaluate social commonsense intelligence, which focuses on understanding people's actions and their social implications.

### E.2 CROSS-LINGUAL SCENARIOS

• **MARC**:The Multilingual Amazon Reviews Corpus (MARC) (Keung et al., 2020) is a large-scale collection of Amazon reviews for multilingual text classification, which contains reviews in six languages: English, Japanese, German, French, Spanish, and Chinese.

# F    DETAILED DESCRIPTION OF BASELINES

In this part, we provide detailed descriptions of all the baselines used in our experiments. Our baselines include prompting methods (*i.e.,* Zero-shot, Few-shot Random, Few-shot TopK (Liu et al., 2022b), Few-shot DPP (Ye et al., 2023)), parameter-efficient fine-tuning methods (*i.e.,* QLoRA (Dettmers et al., 2023) and AdaLoRA (Zhang et al., 2023)), and activation steering methods (*i.e.,* ICV (Liu et al., 2024a) and SEA (Qiu et al., 2024)).

• **Zero-shot**: The model generates predictions using only the input query from the target task, without any demonstrations or examples from source tasks.

• **Few-shot Random**: This method randomly selects a set of examples from the source task.

• **Few-shot TopK** (Liu et al., 2022b): This approach selects examples from the source task based on their similarity to the target input.

• **Few-shot DPP** (Ye et al., 2023): This method uses Determinantal Point Processes (DPP) to select a diverse and representative set of examples from the source task.

• **QLoRA** (Dettmers et al., 2023): Quantized Low-Rank Adaptation (QLoRA) is a parameter-efficient fine-tuning method that uses quantized weights and low-rank adapters to achieve cross-task transfer.

• **AdaLoRA** (Zhang et al., 2023): Adaptive Low-Rank Adaptation (AdaLoRA) dynamically allocates parameter budget based on the importance of different weight matrices. It supports cross-task transfer by fine-tuning on samples from the source task.

• **ICV** (Liu et al., 2024a): In-context Vectors (ICV) steers model behavior by computing and injecting pre-computed PCA-projected activation vectors to the model's hidden states during inference.

• **SEA** (Qiu et al., 2024): Spectral Editing of Activations (SEA) uses SVD to project activations into directions derived from positive and negative demonstrations. It then incorporates the difference representations to steer model behavior toward desired outputs.

# G CASE STUDY

---

**Case Study 1**

**Zero-shot:**
Definition: Given a question answering task from the 3rd to 9th-grade science exam. The question contains four options "A.", "B.", "C." and "D." Select the most appropriate choice that answers the question.
Question: A student mixed 25 grams of salt into 1,000 grams of water. What is the mass of the saltwater mixture?
A. 975 grams
B. 1,000 grams
C. 1,025 grams
D. 2,500 grams
Answer: B

**Few-shot:**
Definition: Given a context and a question do binary true and false type text classification. You are given a passage as context and a question related to the passage that can be answered as "True" or "False". Based on the context, question and your reasoning ability answer in a "True" and "False".
Context: Usually, the relationship between mass and weight on Earth is highly proportional; objects that are a hundred times more massive than a one-liter bottle of soda almost always weigh a hundred times more–approximately 1,000 newtons, which is the weight one would expect on Earth from an object with a mass slightly greater than 100 kilograms. Yet, this is not always the case and there are familiar objects that violate this mass / weight proportionality.
Question: Is mass the same as weight on earth?
Label:False
...

Definition: Given a question answering task from the 3rd to 9th-grade science exam. The question contains four options "A.", "B.", "C." and "D." Select the most appropriate choice that answers the question.
Question: A student mixed 25 grams of salt into 1,000 grams of water. What is the mass of the saltwater mixture?
A. 975 grams
B. 1,000 grams
C. 1,025 grams
D. 2,500 grams
Answer: A

**Ours:**
Definition: Given a question answering task from the 3rd to 9th-grade science exam. The question contains four options "A.", "B.", "C." and "D." Select the most appropriate choice that answers the question.
Question: A student mixed 25 grams of salt into 1,000 grams of water. What is the mass of the saltwater mixture?
A. 975 grams
B. 1,000 grams
C. 1,025 grams
D. 2,500 grams
Answer: C

---

**Case Study 2**

**Zero-shot:**
Definition: Given a sentence mined from a financial news article, you are to determine the sentiment polarity of the sentence. The task deals with financial sentiment analysis. Based on the sentiment conveyed by the sentence, label the sentence as "negative", "positive" or "neutral".
Sentence: Equipment will be manufactured in Vaahto 's workshop in Hollola , Finland and is scheduled for shipments during the first quarter of 2009 .
Label: positive

**Few-shot:**
Definition: Given a context and a question do binary true and false type text classification. You are given a passage as context and a question related to the passage that can be answered as "True" or "False". Based on the context, question and your reasoning ability answer in a "True" and "False".
Context: Harley-Davidson India is a wholly owned subsidiary of Harley-Davidson, based in Gurgaon, Haryana, India. Harley-Davidson India commenced operations in August 2009 and appointed its first dealership in July 2010. Question: does harley davidson have a plant in india Label:True
...

Definition: Given a sentence mined from a financial news article, you are to determine the sentiment polarity of the sentence. The task deals with financial sentiment analysis. Based on the sentiment conveyed by the sentence, label the sentence as "negative", "positive" or "neutral".
Sentence: Equipment will be manufactured in Vaahto 's workshop in Hollola , Finland and is scheduled for shipments during the first quarter of 2009 .
Label: positive

**Ours:**
Definition: Given a sentence mined from a financial news article, you are to determine the sentiment polarity of the sentence. The task deals with financial sentiment analysis. Based on the sentiment conveyed by the sentence, label the sentence as "negative", "positive" or "neutral".
Sentence: Equipment will be manufactured in Vaahto 's workshop in Hollola , Finland and is scheduled for shipments during the first quarter of 2009 .
Label: neutral

*Case Study 3*

**Zero-shot:**
Definition: Given a multiple choice question containing four options "A.", "B.", "C." and "D." from a medical entrance exam. The question is related to a sub-field of medical science like Microbiology, Radiology, Ophthalmology, Surgery, Human anatomy, etc. Based on the question, the option and your knowledge of the medical field select the most appropriate answer from the provided choices "A.", "B.", "C." and "D.".
Question: Which of the following is not a component of quick SOFA (qSOFA) scoring?
A. Bilateral undilated pupils
B. Altered Mentation
C. Glasgow Coma Score
D. SBP <= 100 mm Hg
Answer: C

**Few-shot:**
Definition: Given a context and a question do binary true and false type text classification. You are given a passage as context and a question related to the passage that can be answered as "True" or "False". Based on the context, question and your reasoning ability answer in a "True" and "False".
Context: The series debuted on January 26, 2017 to positive reviews. A 22-episode second season premiered on October 11, 2017, and concluded on May 16, 2018. On April 2, 2018, The CW renewed the series for a third season, which is set to premiere October 10, 2018.
Question: is there going to be any more episodes of riverdale
Label:True
...

Definition: Given a multiple choice question containing four options "A.", "B.", "C." and "D." from a medical entrance exam. The question is related to a sub-field of medical science like Microbiology, Radiology, Ophthalmology, Surgery, Human anatomy, etc. Based on the question, the option and your knowledge of the medical field select the most appropriate answer from the provided choices "A.", "B.", "C." and "D.".
Question: Which of the following is not a component of quick SOFA (qSOFA) scoring?
A. Bilateral undilated pupils
B. Altered Mentation
C. Glasgow Coma Score
D. SBP <= 100 mm Hg
Answer: C

**Ours:**
Definition: Given a multiple choice question containing four options "A.", "B.", "C." and "D." from a medical entrance exam. The question is related to a sub-field of medical science like Microbiology, Radiology, Ophthalmology, Surgery, Human anatomy, etc. Based on the question, the option and your knowledge of the medical field select the most appropriate answer from the provided choices "A.", "B.", "C." and "D.".
Question: Which of the following is not a component of quick SOFA (qSOFA) scoring?
A. Bilateral undilated pupils
B. Altered Mentation
C. Glasgow Coma Score
D. SBP <= 100 mm Hg
Answer: A

*Case Study*

**Zero-shot:**
Definition: Given a question from a scientific exam about Physics, Chemistry, and Biology, among others. The question is in multiple choice format with four answer options "A.", "B.", "C." and "D.". Using your knowledge about the scientific fields answer the question and provide the label "A", "B", "C" and "D" as answer.
Question: What happens to energy when work is done by a system?
A. removed
B. stored
C. multiplied
D. added
Answer: B

**Few-shot:**
Definition: Given a context and a question do binary true and false type text classification. You are given a passage as context and a question related to the passage that can be answered as "True" or "False". Based on the context, question and your reasoning ability answer in a "True" and "False".
Context: The sixth season of the American ABC fantasy-drama Once Upon a Time was ordered on March 3, 2016. It debuted on September 25, 2016, and concluded on May 14, 2017. In January 2017, it was stated that the sixth season would end the main storyline, and for a seventh season, the series would be softly rebooted with a new storyline.
Question: is there a season six of once upon a time
Label:True

...

Definition: Given a question from a scientific exam about Physics, Chemistry, and Biology, among others. The question is in multiple choice format with four answer options "A.", "B.", "C." and "D.". Using your knowledge about the scientific fields answer the question and provide the label "A", "B", "C" and "D" as answer.
Question: What happens to energy when work is done by a system?
A. removed
B. stored
C. multiplied
D. added
Answer: B

**Ours:**
Definition: Given a question from a scientific exam about Physics, Chemistry, and Biology, among others. The question is in multiple choice format with four answer options "A.", "B.", "C." and "D.". Using your knowledge about the scientific fields answer the question and provide the label "A", "B", "C" and "D" as answer.
Question: What happens to energy when work is done by a system?
A. removed
B. stored
C. multiplied
D. added
Answer: A

Table 8: Experiments on cross-domain scenarios using Llama3.1-8B.

| Model | Target Task | Method | Source Task | | | | | | | |
|---|---|---|---|---|---|---|---|---|---|---|
| | | | ARC-Easy | AG-news | BoolQ | Com-QA | MNLI | QQP | SST2 | Average |
| Llama3.1-8B | **ARC-Challenge** (Zs: 71.80) | Few-shot Random | 72.40 | 68.00 | 72.40 | 70.60 | 64.60 | 67.20 | 70.00 | $69.31_{\pm 2.66}$ |
| | | Few-shot TopK | 74.20 | 69.20 | 69.80 | 71.20 | 66.20 | 67.40 | 68.80 | $69.54_{\pm 2.42}$ |
| | | Few-shot DPP | 75.20 | 66.40 | 71.00 | 71.40 | 67.80 | 67.00 | 69.40 | $69.74_{\pm 2.85}$ |
| | | QLoRA | 74.80 | 64.40 | 72.40 | 72.00 | 65.20 | 68.40 | 68.00 | $69.31_{\pm 3.60}$ |
| | | AdaLoRA | 75.00 | 63.80 | 72.80 | 71.80 | 66.40 | 69.20 | 69.20 | $69.74_{\pm 3.55}$ |
| | | ICV | 74.40 | 72.20 | 73.20 | 72.40 | 72.00 | 71.20 | 73.00 | $72.63_{\pm 0.95}$ |
| | | SEA | 74.20 | 72.60 | 73.60 | 74.00 | 73.40 | 72.80 | 73.80 | $73.49_{\pm 0.55}$ |
| | | **FAST** | **75.80** | **76.40** | **75.60** | **77.20** | **76.20** | **75.60** | **76.20** | $\mathbf{76.14}_{\pm 0.52}$ |
| | **Financial Phrasebank** (Zs: 37.80) | Few-shot Random | 44.80 | 48.20 | 46.40 | 48.80 | 56.80 | 48.60 | 46.00 | $48.51_{\pm 3.65}$ |
| | | Few-shot TopK | 42.00 | 47.20 | 46.00 | 47.00 | 61.60 | 52.40 | 44.20 | $48.63_{\pm 6.07}$ |
| | | Few-shot DPP | 40.80 | 47.60 | 46.80 | 49.60 | 61.80 | 49.80 | 45.80 | $48.89_{\pm 5.96}$ |
| | | QLoRA | 44.40 | 54.50 | 52.80 | 49.00 | 63.00 | 56.80 | 45.80 | $52.33_{\pm 6.04}$ |
| | | AdaLoRA | 43.80 | 55.80 | 53.60 | 49.40 | 63.20 | 58.80 | 44.20 | $52.69_{\pm 6.76}$ |
| | | ICV | 48.80 | 56.60 | 55.00 | 51.20 | 61.80 | 58.20 | 50.00 | $54.51_{\pm 4.40}$ |
| | | SEA | 49.60 | 57.80 | 56.20 | 51.40 | 62.40 | 59.00 | 50.60 | $55.29_{\pm 4.49}$ |
| | | **FAST** | **54.60** | **60.40** | **59.40** | **53.40** | **64.00** | **64.40** | **58.60** | $\mathbf{59.26}_{\pm 3.90}$ |
| | **MedMCQA** (Zs: 49.40) | Few-shot Random | 47.00 | 50.00 | 46.80 | 47.60 | 53.20 | 46.60 | 49.60 | $48.69_{\pm 2.23}$ |
| | | Few-shot TopK | 47.80 | 49.00 | 47.80 | 47.40 | 52.80 | 47.40 | 48.60 | $48.69_{\pm 1.77}$ |
| | | Few-shot DPP | 49.00 | 49.80 | 50.20 | 47.40 | 54.00 | 47.20 | 50.60 | $49.74_{\pm 2.12}$ |
| | | QLoRA | 49.20 | 52.00 | 49.20 | 48.40 | 55.20 | 44.20 | 52.20 | $50.06_{\pm 3.24}$ |
| | | AdaLoRA | 49.40 | 52.40 | 49.80 | 48.80 | 55.60 | 44.20 | 51.40 | $50.23_{\pm 3.26}$ |
| | | ICV | 52.60 | 53.60 | 51.00 | 51.80 | 54.20 | 52.20 | 54.00 | $52.77_{\pm 1.11}$ |
| | | SEA | 53.80 | 54.20 | 51.60 | 52.40 | 54.20 | 52.40 | 54.40 | $53.29_{\pm 1.04}$ |
| | | **FAST** | **57.20** | **56.80** | **55.40** | **55.80** | **56.00** | **55.20** | **56.20** | $\mathbf{56.09}_{\pm 0.67}$ |
| | **SciQ** (Zs: 84.40) | Few-shot Random | 88.20 | 84.20 | 86.60 | 87.80 | 80.00 | 81.80 | 83.80 | $84.63_{\pm 2.85}$ |
| | | Few-shot TopK | 87.60 | 82.20 | 85.80 | 87.00 | 83.20 | 81.60 | 86.60 | $84.86_{\pm 2.28}$ |
| | | Few-shot DPP | 87.40 | 86.40 | 87.00 | 87.80 | 82.60 | 82.00 | 83.60 | $85.26_{\pm 2.26}$ |
| | | QLoRA | 88.60 | 83.00 | 88.80 | 88.00 | 77.00 | 78.20 | 84.20 | $83.97_{\pm 4.53}$ |
| | | AdaLoRA | 88.40 | 82.80 | 88.60 | 87.80 | 78.20 | 78.40 | 85.30 | $84.21_{\pm 4.19}$ |
| | | ICV | 89.80 | 89.20 | 90.20 | 88.20 | 84.00 | 83.80 | 87.00 | $87.46_{\pm 2.45}$ |
| | | SEA | 90.20 | 89.40 | 90.60 | 88.40 | 85.20 | 84.80 | 87.40 | $88.00_{\pm 2.14}$ |
| | | **FAST** | **91.60** | **92.00** | **91.00** | **90.00** | **89.00** | **89.80** | **89.60** | $\mathbf{90.43}_{\pm 1.03}$ |
| | **Social-i-QA** (Zs: 55.40) | Few-shot Random | 60.40 | 58.00 | 59.60 | 62.40 | 59.00 | 57.20 | 57.40 | $59.14_{\pm 1.71}$ |
| | | Few-shot TopK | 61.80 | 60.80 | 58.80 | 62.80 | 58.60 | 56.80 | 56.00 | $59.37_{\pm 2.35}$ |
| | | Few-shot DPP | 60.20 | 61.00 | 60.00 | 60.00 | 60.00 | 59.60 | 56.40 | $59.60_{\pm 1.36}$ |
| | | QLoRA | 61.40 | 56.00 | 61.00 | 64.40 | 55.20 | 54.20 | 52.20 | $57.77_{\pm 4.16}$ |
| | | AdaLoRA | 62.00 | 56.40 | 60.60 | 65.40 | 56.00 | 53.80 | 51.00 | $57.89_{\pm 4.64}$ |
| | | ICV | 63.00 | 61.40 | 61.00 | 64.80 | 60.20 | 60.20 | 60.20 | $61.54_{\pm 1.62}$ |
| | | SEA | 62.80 | 61.80 | 61.80 | 64.80 | 61.40 | 61.80 | 60.40 | $62.11_{\pm 1.28}$ |
| | | **FAST** | **65.60** | **63.20** | **64.20** | **66.40** | **63.40** | **65.60** | **63.60** | $\mathbf{64.57}_{\pm 1.18}$ |

Table 9: Experiments on cross-domain scenarios using Qwen2.5-7B.

| Model | Target Task | Method | Source Task | | | | | | | |
|---|---|---|---|---|---|---|---|---|---|---|
| | | | ARC-Easy | AG-news | BoolQ | Com-QA | MNLI | QQP | SST2 | Average |
| Qwen2.5-7B | **ARC-Challenge** (Zs: 82.80) | Few-shot Random | 86.80 | 86.60 | 83.20 | 86.60 | 84.40 | 86.20 | 86.80 | $85.80_{\pm1.32}$ |
| | | Few-shot TopK | 87.40 | 86.40 | 86.00 | 86.60 | 84.60 | 86.60 | 86.60 | $86.31_{\pm0.80}$ |
| | | Few-shot DPP | 87.20 | 86.60 | 86.00 | 86.80 | 85.00 | 86.00 | 86.60 | $86.31_{\pm0.67}$ |
| | | QLoRA | 91.20 | 82.20 | 84.00 | 87.80 | 82.20 | 87.20 | 89.80 | $86.34_{\pm3.34}$ |
| | | AdaLoRA | 91.40 | 83.40 | 82.60 | 87.60 | 81.20 | 87.60 | 90.40 | $86.31_{\pm3.67}$ |
| | | ICV | 90.80 | 88.00 | 88.20 | 88.00 | 87.40 | 89.00 | 88.00 | $88.49_{\pm1.04}$ |
| | | SEA | 91.20 | 89.40 | 89.00 | 88.40 | 87.80 | 89.40 | 88.80 | $89.14_{\pm0.99}$ |
| | | **FAST** | **93.40** | **92.20** | **92.40** | **92.80** | **90.00** | **91.80** | **92.80** | $\mathbf{92.20_{\pm1.01}}$ |
| | **Financial Phrasebank** (Zs: 85.20) | Few-shot Random | 87.60 | 86.80 | 89.80 | 87.00 | 85.60 | 84.00 | 87.40 | $86.89_{\pm1.66}$ |
| | | Few-shot TopK | 89.20 | 88.00 | 89.60 | 86.40 | 84.00 | 86.60 | 90.00 | $87.69_{\pm1.99}$ |
| | | Few-shot DPP | 87.60 | 87.80 | 90.40 | 86.20 | 86.20 | 86.80 | 89.20 | $87.74_{\pm1.46}$ |
| | | QLoRA | 89.20 | 84.20 | 92.40 | 82.20 | 84.80 | 83.80 | 92.80 | $87.06_{\pm4.03}$ |
| | | AdaLoRA | 89.40 | 83.80 | 93.60 | 83.60 | 85.20 | 85.20 | 93.20 | $87.71_{\pm4.01}$ |
| | | ICV | 90.20 | 89.80 | 92.80 | 89.60 | 88.00 | 89.20 | 93.40 | $90.43_{\pm1.81}$ |
| | | SEA | 90.80 | 90.40 | 93.60 | 89.80 | 88.60 | 89.00 | 93.00 | $90.74_{\pm1.77}$ |
| | | **FAST** | **94.60** | **93.20** | **94.60** | **93.80** | **92.60** | **93.60** | **94.20** | $\mathbf{93.80_{\pm0.68}}$ |
| | **MedMCQA** (Zs: 52.00) | Few-shot Random | 54.80 | 53.60 | 55.80 | 55.40 | 52.00 | 55.00 | 54.00 | $54.37_{\pm1.20}$ |
| | | Few-shot TopK | 54.80 | 54.20 | 55.40 | 55.00 | 53.00 | 54.80 | 54.60 | $54.54_{\pm0.71}$ |
| | | Few-shot DPP | 55.60 | 53.40 | 54.20 | 55.00 | 53.80 | 55.40 | 54.80 | $54.60_{\pm0.76}$ |
| | | QLoRA | 56.20 | 51.20 | 54.20 | 60.40 | 50.20 | 60.40 | 56.00 | $55.51_{\pm3.72}$ |
| | | AdaLoRA | 56.40 | 50.80 | 54.80 | 60.80 | 50.80 | 61.20 | 56.20 | $55.86_{\pm3.89}$ |
| | | ICV | 59.20 | 55.00 | 57.40 | 58.60 | 55.80 | 58.20 | 57.20 | $57.34_{\pm1.40}$ |
| | | SEA | 59.60 | 56.20 | 57.80 | 60.20 | 56.40 | 59.40 | 57.20 | $58.11_{\pm1.50}$ |
| | | **FAST** | **62.00** | **59.00** | **60.60** | **63.00** | **60.00** | **62.40** | **59.00** | $\mathbf{60.86_{\pm1.51}}$ |
| | **SciQ** (Zs: 89.60) | Few-shot Random | 91.20 | 89.60 | 91.60 | 91.00 | 88.40 | 88.20 | 87.40 | $89.63_{\pm1.55}$ |
| | | Few-shot TopK | 91.20 | 89.20 | 92.20 | 91.20 | 88.80 | 88.80 | 88.80 | $90.03_{\pm1.35}$ |
| | | Few-shot DPP | 91.40 | 89.00 | 92.00 | 91.60 | 88.60 | 89.80 | 87.20 | $89.94_{\pm1.66}$ |
| | | QLoRA | 93.20 | 86.20 | 94.60 | 93.80 | 85.20 | 84.60 | 83.40 | $88.71_{\pm4.54}$ |
| | | AdaLoRA | 93.80 | 87.40 | 94.60 | 94.40 | 86.40 | 85.20 | 84.60 | $89.49_{\pm4.23}$ |
| | | ICV | 93.80 | 90.00 | 93.60 | 93.60 | 90.00 | 91.20 | 91.80 | $91.91_{\pm1.48}$ |
| | | SEA | 94.40 | 91.60 | 93.40 | 94.00 | 91.00 | 91.80 | 92.40 | $92.66_{\pm1.20}$ |
| | | **FAST** | **96.40** | **94.20** | **95.80** | **95.40** | **94.80** | **95.00** | **95.00** | $\mathbf{95.23_{\pm0.66}}$ |
| | **Social-i-QA** (Zs: 76.00) | Few-shot Random | 78.60 | 75.60 | 78.40 | 77.00 | 78.00 | 76.00 | 75.40 | $77.00_{\pm1.26}$ |
| | | Few-shot TopK | 78.00 | 76.20 | 77.60 | 77.20 | 78.00 | 76.00 | 75.80 | $76.97_{\pm0.88}$ |
| | | Few-shot DPP | 78.00 | 75.40 | 78.80 | 76.60 | 77.20 | 75.60 | 76.20 | $76.83_{\pm1.16}$ |
| | | QLoRA | 81.20 | 73.20 | 79.20 | 80.40 | 81.40 | 72.40 | 73.40 | $77.31_{\pm3.80}$ |
| | | AdaLoRA | 81.80 | 73.00 | 79.80 | 80.20 | 81.40 | 73.20 | 73.80 | $77.60_{\pm3.75}$ |
| | | ICV | 81.60 | 80.20 | 79.80 | 81.20 | 80.80 | 78.00 | 79.20 | $80.11_{\pm1.15}$ |
| | | SEA | 82.00 | 80.80 | 80.20 | 81.60 | 81.60 | 78.00 | 79.00 | $80.46_{\pm1.38}$ |
| | | **FAST** | **84.20** | **83.80** | **83.60** | **84.40** | **83.80** | **81.80** | **82.60** | $\mathbf{83.46_{\pm0.86}}$ |

Table 10: Performance comparison in the cross-lingual scenarios.

| Target Language | Method | Source Language | | | | | | |
|---|---|---|---|---|---|---|---|---|
| | | de | en | es | fr | ja | zh | Average |
| **de** (Zs: 84.40) | Few-shot Random | - | 76.80 | 84.00 | 93.00 | 87.60 | 86.80 | $85.64_{\pm 5.30}$ |
| | Few-shot TopK | - | 77.20 | 85.40 | 93.80 | 86.80 | 87.40 | $86.12_{\pm 5.31}$ |
| | Few-shot DPP | - | 77.20 | 86.20 | 92.60 | 87.20 | 88.60 | $86.36_{\pm 5.07}$ |
| | QLoRA | - | 75.20 | 88.40 | 94.20 | 83.20 | 84.80 | $85.16_{\pm 6.25}$ |
| | AdaLoRA | - | 75.80 | 88.00 | 94.60 | 84.40 | 85.40 | $85.64_{\pm 6.07}$ |
| | ICV | - | 86.60 | 88.20 | 94.00 | 91.40 | 90.60 | $90.16_{\pm 2.57}$ |
| | SEA | - | 87.00 | 88.60 | 94.80 | 91.00 | 90.80 | $90.44_{\pm 2.63}$ |
| | **FAST** | - | **89.80** | **90.80** | **95.80** | **93.20** | **92.80** | $\mathbf{92.48}_{\pm 2.08}$ |
| **en** (Zs: 66.00) | Few-shot Random | 77.40 | - | 42.80 | 58.00 | 76.80 | 41.80 | $59.36_{\pm 15.58}$ |
| | Few-shot TopK | 76.80 | - | 51.00 | 62.40 | 79.60 | 38.00 | $61.56_{\pm 15.65}$ |
| | Few-shot DPP | 76.80 | - | 58.80 | 68.80 | 80.00 | 38.00 | $64.48_{\pm 15.13}$ |
| | QLoRA | 71.40 | - | 60.60 | 72.80 | 85.80 | 26.80 | $63.48_{\pm 20.01}$ |
| | AdaLoRA | 73.60 | - | 62.40 | 74.60 | 86.40 | 28.40 | $65.08_{\pm 19.85}$ |
| | ICV | 85.40 | - | 80.40 | 85.80 | 86.20 | 76.20 | $82.80_{\pm 3.92}$ |
| | SEA | 86.80 | - | 81.60 | 87.40 | 85.20 | 79.80 | $84.16_{\pm 2.97}$ |
| | **FAST** | **90.60** | - | **86.00** | **91.20** | **90.40** | **87.00** | $\mathbf{89.04}_{\pm 2.11}$ |
| **es** (Zs: 81.60) | Few-shot Random | 83.80 | 83.80 | - | 83.60 | 81.60 | 82.20 | $83.48_{\pm 9.58}$ |
| | Few-shot TopK | 84.00 | 83.00 | - | 84.20 | 83.00 | 83.20 | $83.68_{\pm 9.73}$ |
| | Few-shot DPP | 84.00 | 84.20 | - | 83.60 | 83.40 | 83.40 | $85.40_{\pm 8.42}$ |
| | QLoRA | 87.80 | 85.20 | - | 84.60 | 81.40 | 74.80 | $82.16_{\pm 13.17}$ |
| | AdaLoRA | 88.60 | 84.40 | - | 84.20 | 80.20 | 76.80 | $83.00_{\pm 14.07}$ |
| | ICV | 92.20 | 91.60 | - | 92.40 | 91.20 | 89.40 | $91.96_{\pm 2.06}$ |
| | SEA | 91.40 | 92.40 | - | 93.20 | 92.00 | 89.60 | $92.20_{\pm 1.62}$ |
| | **FAST** | **95.20** | **95.60** | - | **95.20** | **94.60** | **93.80** | $\mathbf{95.20}_{\pm 0.40}$ |
| **fr** (Zs: 86.60) | Few-shot Random | 91.80 | 65.60 | 84.00 | - | 91.80 | 84.20 | $81.40_{\pm 9.65}$ |
| | Few-shot TopK | 91.40 | 65.40 | 86.80 | - | 92.00 | 82.80 | $81.75_{\pm 9.99}$ |
| | Few-shot DPP | 91.40 | 69.40 | 89.40 | - | 92.20 | 84.60 | $83.90_{\pm 8.80}$ |
| | QLoRA | 93.40 | 58.20 | 88.00 | - | 93.00 | 78.20 | $79.35_{\pm 13.32}$ |
| | AdaLoRA | 94.20 | 56.80 | 90.20 | - | 93.80 | 80.00 | $80.20_{\pm 14.43}$ |
| | ICV | 93.00 | 91.40 | 92.40 | - | 94.60 | 88.40 | $91.70_{\pm 2.23}$ |
| | SEA | 93.60 | 92.00 | 91.60 | - | 94.20 | 89.60 | $91.85_{\pm 1.63}$ |
| | **FAST** | **95.00** | **95.20** | **95.80** | - | **95.40** | **94.60** | $\mathbf{95.25}_{\pm 0.43}$ |
| **ja** (Zs: 38.60) | Few-shot Random | 49.00 | 27.80 | 40.60 | 43.00 | - | 36.00 | $39.28_{\pm 7.11}$ |
| | Few-shot TopK | 49.40 | 26.00 | 41.40 | 43.60 | - | 36.60 | $39.40_{\pm 7.86}$ |
| | Few-shot DPP | 50.40 | 25.20 | 43.40 | 44.00 | - | 36.20 | $39.84_{\pm 8.59}$ |
| | QLoRA | 50.40 | 15.60 | 45.60 | 46.80 | - | 30.20 | $37.72_{\pm 13.05}$ |
| | AdaLoRA | 50.80 | 18.40 | 46.00 | 46.40 | - | 32.40 | $38.80_{\pm 11.92}$ |
| | ICV | 51.20 | 40.40 | 44.80 | 45.40 | - | 39.80 | $44.32_{\pm 4.11}$ |
| | SEA | 51.80 | 42.00 | 45.40 | 44.80 | - | 41.40 | $45.08_{\pm 3.70}$ |
| | **FAST** | **53.20** | **47.00** | **47.60** | **47.00** | - | **45.80** | $\mathbf{48.12}_{\pm 2.61}$ |
| **zh** (Zs: 30.80) | Few-shot Random | 49.00 | 35.00 | 33.60 | 32.00 | 37.20 | - | $37.36_{\pm 6.07}$ |
| | Few-shot TopK | 49.20 | 36.60 | 35.80 | 33.20 | 37.00 | - | $38.36_{\pm 5.58}$ |
| | Few-shot DPP | 51.60 | 35.80 | 37.00 | 31.40 | 37.80 | - | $38.72_{\pm 6.81}$ |
| | QLoRA | 51.20 | 28.60 | 34.00 | 26.80 | 39.00 | - | $35.92_{\pm 8.76}$ |
| | AdaLoRA | 52.40 | 28.20 | 35.20 | 28.40 | 39.80 | - | $36.80_{\pm 8.94}$ |
| | ICV | 52.00 | 38.00 | 39.00 | 36.60 | 40.20 | - | $41.16_{\pm 5.55}$ |
| | SEA | 53.20 | 39.20 | 38.80 | 38.80 | 41.20 | - | $42.24_{\pm 5.55}$ |
| | **FAST** | **53.20** | **42.80** | **42.20** | **43.20** | **43.00** | - | $\mathbf{44.88}_{\pm 4.17}$ |

