# OpenReview forum: "Enhancing Cross-task Transfer of Large Language Models via Fourier Activation Steering"
_ICLR.cc/2026/Conference — ICLR 2026 Conference Withdrawn Submission_

### Official Review · Reviewer_gpcn · 2025-10-31

**Soundness:** 3
**Presentation:** 2
**Contribution:** 3
**Rating:** 4
**Confidence:** 3

**Summary:**

This paper proposes a latent space steering method for cross-task transfer learning in LLMs, enabling knowledge from high-resource tasks to enhance performance on low-resource tasks. The method is built on an analysis of how In-contest learning (ICL) functions at the level of model activations.

The paper first establishes that ICL (providing few-shot examples) works by “shifting” the model’s internal activations. This makes a distinct difference vector (dv) between the activations of a zero-shot prompt and a few-shot prompt. This implies that the learning from examples can be captured as a directional vector in the model’s latent space.

To extract task-agnostic information from these vectors dv the authors apply FFT. This transformation decomposes the vectors into low/high frequency components. The low-frequency component is found to be task-agnostic, while the high-frequency component is task-specific.

The overall proposed method consists of two parts: first, selecting an influential and diverse subset of samples from the high-resource source task via a graph-based algorithm to compute the difference vector dv; and second, injecting this vector's components into the latent space, using Fourier decomposition to filter and separate them.

**Strengths:**

I think the paper’s key contribution is its novelty of the approach.

It begins by empirically identifying a core phenomenon, ICL as a consistent activation shift. It then defines the challenge this presents (the shift is task-specific, potentially leading to negative transfer).

It treats the complex and high-dimensional activation vectors as “signals” and applies Fourier transform. This leads to a compelling and insightful interpretation: the low-frequency components corresponds to task-agnostic information, while the high-frequency components encode task-specific details.

**Weaknesses:**

- The choice of FFT: Although FFT provides a good starting point, there are more advanced methods in signal processing for more sophisticated analysis, e.g. wavelets.

- Convoluted method: The proposed method is complex and relies on several key hyperparameters that could be difficult to tune. The performance is likely sensitive to:
The frequency cutoff $k$ (Eq. 5) used to separate low and high frequencies.
The injection strength $\lambda$ (Eq. 10)
The similarity threshold $\epsilon$ (Eq. 10)
and The parameters for the graph-based sampling (e.g., number of neighbors, diffusion steps).
These settings may not generalize well across different models or tasks, requiring careful tuning for each new application.

**Questions:**

(see weaknesses)

---

> ### Author Response · Authors · 2025-11-20
> **Rebuttal by Authors (1/2)**
>
> ### **W1: The choice of FFT: Although FFT provides a good starting point, there are more advanced methods in signal processing for more sophisticated analysis, e.g. wavelets.**
>
> We appreciate the reviewer's insightful suggestion regarding advanced signal processing methods.
> We agree that FFT serves as a solid starting point, and to our knowledge, which is the **first** work to systematically analyze cross-task transfer through **spectral decomposition**.
>
> Following the reviewer's suggestion, we conduct additional experiments using wavelet transforms [1] and Discrete Cosine Transform [2] to extract low- and high-frequency components, which were then injected using our same activation steering framework.
> The results are as follows:
>
>
> | Dataset  | ARC-C | FPB | MedMCQA |
> | -------- | --------  | --------  | -------- |
> | **FFT**                   | **76.14** | **59.26** | 56.09    |
> | Wavelets Transform        | 75.89 | 58.82 | 55.82 |
> | Discrete Cosine Transform | 76.08 | 59.23 | **56.30** |
>
> We observe robust performance across different signal processing methods, with minimal variation in the results, which supports the **generalizability** of our core finding: low-frequency components capture task-agnostic enhancement signals, while high-frequency components encode task-specific representations.
> It further confirms the **effectiveness and robustness** of our proposed approach.
>
> [1] A theory for multiresolution signal decomposition: the wavelet representation, TPAMI.
>
> [2] Discrete Cosine Transform. IEEE Transactions on Computers.

---

> ### Author Response · Authors · 2025-11-20
> **Rebuttal by Authors (2/2)**
>
> ### **W2: Convoluted method: The proposed method is complex and relies on several key hyperparameters that could be difficult to tune. The performance is likely sensitive to: The frequency cutoff $k$ (Eq. 5) used to separate low and high frequencies. The injection strength $\lambda$ (Eq. 10) The similarity threshold $\epsilon$ (Eq. 10) and The parameters for the graph-based sampling (e.g., number of neighbors, diffusion steps). These settings may not generalize well across different models or tasks, requiring careful tuning for each new application.**
>
> We thank the reviewer for raising these important points regarding hyperparameter sensitivity.
> We have conducted comprehensive experiments and found that our method is generally robust to these hyperparameters.
>
> (1) **Graph-based sampling parameters:**
>
> + **Number of neighbors**: Following Vote-k [1], we set this to 150. Our analysis shows minimal sensitivity around this value, and the performance remains stable between 100 and 200 neighbors.
>
> | Neighbor Numbers | 50 | 100 | 150 | 200 | 250 |
> | -------- | -------- | -------- | -------- | -------- | -------- |
> | ARC-Challenge            | 75.77 | 76.08 | **76.14** | 76.09 | 75.92 |
> | Financial Phrasebank     | 58.94 | **59.33** | 59.26 | 59.19 | 58.62 |
>
> + **Diffusion steps**: We follow IDEAL [2] and set this to 10. While more steps slightly improve results, we balance performance and efficiency.
>
> | Diffusion process times | 5 | 10 | 15 | 20 |
> | -------- | -------- | -------- | -------- | -------- |
> | ARC-Challenge            | 75.28 | **76.14** | 76.20 | 76.34 |
> | Financial Phrasebank     | 58.47 | **59.26** | 59.48 | 59.51 |
>
> (2) **activation steering parameters:**
>
> + **Injection strength $\lambda$ (Eq. 10):** As shown in **Figure 8** (with additional results in **Figure 11**), we observe that injection strengths between 0.1 and 1 **consistently** improve performance over the zero-shot baseline (71.80). The optimal value is 0.2 (achieving 76.14), while even λ=1 yields 74.74, which still has a significant gain over zero-shot.
>
> + **Frequency cutoff $k$**: We find that the frequency cutoff $k=d/2$ yields the best performance, as it effectively separates low- and high-frequency components.
>
>
>
> | Frequency cutoff | $d/4$ | $d/3$ | $d/2$ | $2d/3$ |
> | -------- | -------- | -------- | -------- | -------- |
> | ARC-Challenge            | 73.95 | 75.54 | **76.14** | 75.23 |
> | Financial Phrasebank     | 57.33 | 58.57 | **59.26** | 58.12 |
>
>
>
>
> + **Similarity threshold $\epsilon$:** The optimal similarity threshold is 0.6. Performance remains relatively stable across values from 0.4 to 0.7, with our method consistently outperforming all baselines in this range.
>
> | Similarity threshold | 0.4 | 0.5 | 0.6 | 0.7 |
> | -------- | -------- | -------- | -------- | -------- |
> | ARC-Challenge            | 75.12 | 75.93 | **76.14** | 75.84 |
> | Financial Phrasebank     | 58.25 | 59.21 | **59.26** | 58.98 |
>
> Across all parameters, our method outperforms baselines within reasonable ranges, demonstrating practical **robustness**.
>
> We have included the hyperparameter analysis of frequency cutoff $k$ and similarity threshold $\epsilon$ in **Appendix D.1**.
>
> [1] Selective Annotation Makes Language Models Better Few-Shot Learners, ICLR 2023.
>
> [2] IDEAL: Influence-Driven Selective Annotations Empower In-Context Learners in Large Language Models, ICLR 2024.

---

> ### Author Response · Authors · 2025-11-27
> **Kindly request your feedback before the end of the discussion period**
>
> Dear Reviewer gpcn:
>
> As the author-reviewer discussion period is soon ending, we would appreciate it if you could review our responses and provide your feedback at your earliest convenience. If there are any further questions or comments, we will do our best to address them before the discussion period ends.
>
> Thank you very much for your time and efforts!
>
> Sincerely,
>
> Authors

---

> > ### Comment · Reviewer_gpcn · 2025-11-27
> > **additional results appreciated**
> >
> > Thank you authors for all the hard work gathering additional results. I highly value the main motivation and the novel insight behind the approach. It's just that its execution involving multiple steps leaves room for improvement. We all aspire simpler training paradigms for innovation. In this regards, I will maintain my score.

---

> > > ### Author Response · Authors · 2025-11-27
> > >
> > > Thank you for taking the time to review our response and for acknowledging the **main motivation**, **novel insights**, and **solid experiments** in our work.
> > > We truly appreciate your positive recognition of these aspects.
> > >
> > > We would like to reiterate the core contributions of our paper:
> > >
> > > Our analysis is structured into three key components:
> > > + **Section 2.2** demonstrates that information-enhanced features induced by in-context examples exhibit consistent patterns across tasks.
> > > + **Section 2.3** reveals that the difference activation directions arising from these samples are highly correlated with task similarity.
> > > + **Section 2.4** employs Fourier analysis to decompose activation directions, showing that low-frequency components encode information-enhanced features, while high-frequency components retain task-specific information.
> > >
> > > To the best of our knowledge, our work is the **first** to systematically analyze cross-task activations in this manner.
> > > The insights we provide are **valuable** and have the potential to inspire and guide future research in representation engineering.
> > >
> > > Methodologically, our approach consists of two stages:
> > > + **Influence and diverse subset selection**: This step ensures the quality of source task data by selecting a representative subset from a large labeled pool.
> > > + **Fourier-based Activation Steering**: This enhances forward inference by incorporating low-frequency, information-rich features and selectively integrating task-specific representations.
> > >
> > > This two-stage pipeline enables cross-task transfer without requiring architectural modifications or extra supervision, making it both **practical and accessible** for real-world applications.
> > >
> > > We also note that recent studies in representation engineering [1,2,3,4] have **gained significant recognition** in the community, further validating the importance of our research direction.
> > >
> > > We firmly believe that high-impact research lies **not only in simplicity but also in the novelty, depth, and completeness of the insights provided**.
> > > Our work offers a fresh perspective and a comprehensive analysis, so we are confident that it will contribute meaningfully to the community.
> > >
> > > We are happy to engage in further discussion regarding any aspect of the paper and welcome the opportunity to address any additional questions you may have.
> > >
> > > [1] Improving Reasoning Performance in Large Language Models via Representation Engineering, ICLR 2025.
> > >
> > > [2] Adversarial Representation Engineering: A General Model Editing Framework for Large Language Models, NIPS 2024.
> > >
> > > [3] Improving Alignment and Robustness with Circuit Breakers, NIPS 2024.
> > >
> > > [4] Improving Instruction-Following in Language Models through Activation Steering, ICLR 2025.

---

### Official Review · Reviewer_8p6w · 2025-11-01

**Soundness:** 3
**Presentation:** 2
**Contribution:** 3
**Rating:** 6
**Confidence:** 3

**Summary:**

This paper introduces FAST (Fourier-based Activation Steering for cross-task Transfer), a framework that enables large language models (LLMs) to transfer knowledge from high-resource to low-resource tasks without fine-tuning or input expansion. The authors first observe that the difference between few-shot and zero-shot activations forms nearly parallel patterns across tasks and that these differences correlate with task similarity. Using Fourier analysis, they decompose these activations into low-frequency components that encode task-agnostic, information-enhanced features, and high-frequency components that capture task-specific details. FAST selects a diverse and influential subset of high-resource samples, extracts their activation differences, applies Fourier filtering, and injects the resulting components into the target model during inference to steer behavior. Experiments across cross-domain and cross-lingual transfer settings show that FAST consistently outperforms prompting, fine-tuning, and other activation-steering baselines while maintaining efficiency and scalability.

**Strengths:**

- The paper introduces a novel Fourier-based approach to activation steering that enables cross-task transfer without fine-tuning or retraining, offering a conceptually elegant and computationally lightweight framework.

- The method demonstrates consistent performance gains across multiple cross-domain and cross-lingual benchmarks, which shows broad applicability and robustness.

- The empirical analysis provides strong evidence that activation patterns encode transferable task information, which offers new insight into how internal representations can be reused across tasks.

- The approach integrates smoothly with existing LLMs and inference pipelines, requiring no architectural modification or additional supervision, which increases its practical utility.

- The visualization and qualitative analyses effectively illustrate how Fourier filtering separates task-agnostic and task-specific components, supporting the interpretability of the proposed method.

**Weaknesses:**

- The method's theoretical justification is limited. The claim that low-frequency components capture task-agnostic semantics is supported only by empirical correlations -- specifically, the similarity analysis in Figure 3 and the performance comparisons in Figure 4, rather than quantitative causal ablations or theoretical derivations.
- The sample selection strategy lacks clarity and robustness. The paper introduces "diverse and influential" sample selection (Section 3.3) but provides no sensitivity analysis or comparison to random or simpler baselines, which weakens its empirical foundation.
- The evaluation is narrow in scope. Most experiments focus on a few NLU (Table 1) and translation datasets (Table 2), leaving open whether FAST generalizes to reasoning or multimodal adaptation.
- The computational overhead is underexplored. Though the authors claim efficiency, they omit runtime or memory statistics for Fourier transforms and activation steering across layers, which makes the scalability claim unsubstantiated.
- The ablation studies are insufficient to isolate design contributions. Table 3 reports limited variations, but it does not disentangle the effects of Fourier filtering, activation injection depth, and steering magnitude, so the relative importance of each component remains unclear.

**Questions:**

How sensitive is FAST to the choice of frequency cutoff when separating low- and high-frequency components, and how should practitioners select this threshold for new tasks or models?

---

> ### Author Response · Authors · 2025-11-20
> **Rebuttal by Authors (1/3)**
>
> Thank you very much for the time and effort you have dedicated to reviewing our paper.
> We sincerely appreciate your insightful comments and are pleased to address all of your concerns.
>
> In the following part, we provide responses to each of the questions you raised.
> We have also incorporated your suggestions into the revised PDF, with all corresponding changes **highlighted in green** for your easy reference.
>
>
>
> ### **W1: The method's theoretical justification is limited. The claim that low-frequency components capture task-agnostic semantics is supported only by empirical correlations -- specifically, the similarity analysis in Figure 3 and the performance comparisons in Figure 4, rather than quantitative causal ablations or theoretical derivations.**
>
> We agree that providing theoretical derivations or causal validation for the link between low-frequency components and task-agnostic semantics is an important and valuable direction.
>
> We would like to clarify that our study follows the research paradigm of representation engineering, which is a top-down approach to transparency research that treats representations as the fundamental unit of analysis, with the goal of understanding and controlling representations of high-level cognitive phenomena in neural networks through systematic empirical analysis [1,2].
> **At this stage, strong empirical correlations are essential for establishing the foundation of such phenomena.**
>
> In our work, the similarity analysis in Figure 3 reveals a **structural phenomenon**: low-frequency representations remain stable across different tasks. The performance comparisons in Figure 4 further demonstrate that manipulating this structural feature leads to systematic changes in model behavior.
> The hypothesis that **low-frequency components capture task-agnostic semantic-enhancing information** provides a coherent explanation for both empirical observations.
>
> Therefore, we believe that **strong correlational evidence is not only appropriate but necessary in the current phase for this field**.
> Additionally, our work contributes a reliable empirical foundation for future theoretical formalization and causal testing.
>
> ### **W2: The sample selection strategy lacks clarity and robustness. The paper introduces "diverse and influential" sample selection (Section 3.3) but provides no sensitivity analysis or comparison to random or simpler baselines, which weakens its empirical foundation.**
>
> Actually, we already conducted ablation studies in **Table 3**, where we evaluated the impact of removing the influence score, the diversity penalty, and both components.
> Here, **removing both is equivalent to random subset selection**.
> As the reviewer suggested, we have now added comparisons with two additional subset selection methods: Vote-k [1] and IDEAL [2].
> The updated results are as follows:
>
> | Dataset  | ARC-C | FPB | MedMCQA |
> | -------- | --------  | --------  | -------- |
> | **FAST**     | **76.14**     | **59.26**     | **56.09**    |
> | w/o Influence score   | 75.62  | 56.91       | 54.83        |
> | w/o Diversity Penalty | 74.91  | 57.38       | 55.29        |
> | w/o Both(Random)      | 73.72  | 54.29       | 53.18        |
> | Vote-k                | 75.46  | 57.89       | 55.35        |
> | IDEAL                 | 75.69  | 58.04       | 55.04        |
>
> The results show that removing either the influence score or diversity penalty leads to performance degradation, confirming that **both components contribute to our method**.
> Moreover, our selection strategy **outperforms** both Vote-k and IDEAL, further validating the effectiveness of our proposed approach.
>
> We have included these two baselines in **Table 3** and added corresponding discussions in **Section 4.3**.
>
> [1] Selective Annotation Makes Language Models Better Few-Shot Learners, ICLR 2023.
>
> [2] IDEAL: Influence-Driven Selective Annotations Empower In-Context Learners in Large Language Models, ICLR 2024.

---

> ### Author Response · Authors · 2025-11-20
> **Rebuttal by Authors (2/3)**
>
> ### **W3: The evaluation is narrow in scope. Most experiments focus on a few NLU (Table 1) and translation datasets (Table 2), leaving open whether FAST generalizes to reasoning or multimodal adaptation.**
>
> We thank the reviewer for this important point regarding the scope of evaluation. Assessing generalization to reasoning and multimodal tasks is indeed crucial for validating the versatility of our method.
> In fact, we had already conducted experiments on four generation benchmarks in **Appendix D.1** (now in **Section 4.5**):
> + XSum for summarization
> + GSM8K for mathematical reasoning
> + GPQA for scientific question answering
> + LiveCodeBench for code generation.
>
> The latter three tasks involve **mathematical, STEM, and coding domains**, which are datasets for evaluating reasoning capabilities.
> As the reviewer suggested, we have now moved these experiments to the main body.
>
> Additionally, we have conducted experiments on multimodal tasks, including the MathVista, MMStar, and MMMU datasets, using Qwen2.5-VL-7B.
> The results are as follows:
>
> | Method          | MathVista | MMStar | MMMU | Average |
> | --------        | --------  | --------  | --------  | --------  |
> | Zero-shot       | 68.52    | 64.12  | 58.08 | 63.57 |
> | Few-shot Random | 64.43    | 62.12  | 59.71 | 62.09 |
> | Few-shot TopK   | 64.48    | 62.28  | 59.23 | 62.00 |
> | Few-shot DPP    | 64.91    | 62.54  | 60.18 | 62.54 |
> | QLoRA           | 60.53    | 65.23  | 58.24 | 61.33 |
> | AdaLoRA         | 62.91    | 65.26  | 58.46 | 62.21 |
> | ICV             | 71.17    | 65.19  | 61.24 | 65.87 |
> | SEA             | 71.24    | 66.50  | 61.85 | 66.53 |
> | **Ours**        | **73.34**| **68.83**  | **64.12** | **68.76** |
>
>
> The results show that our method consistently achieves the best performance on these multimodal reasoning tasks, demonstrating its broad applicability.
>
> These additional experiments have been added to **Appendix D.2** in the revised version.
>
>
> ### **W4: The computational overhead is underexplored. Though the authors claim efficiency, they omit runtime or memory statistics for Fourier transforms and activation steering across layers, which makes the scalability claim unsubstantiated.**
>
> We thank the reviewer for raising this important point about computational overhead.
> In fact, we have already provided a detailed analysis of computational costs in **Appendix D.2** (now in **Section 4.4**), including **time complexity** and **measured runtime for preprocessing, training, and inference process**.
> As shown in **Table 5** (now in **Table 4**), the total time required by our method is 393 seconds, which includes 172 seconds for preprocessing (subset selection and activation extraction) and 221 seconds for inference.
> Within the preprocessing stage, both the Fourier transform and activation steering (i.e., element-wise addition across layers) contribute **minimal** time, which requires only **less than 1 second** in total.
> This demonstrates a notable advantage over both few-shot and PEFT methods, confirming that our approach achieves effective cross-task transfer while maintaining high computational efficiency.
>
> Following the reviewer's suggestion, we have moved this analysis to **Section 4.4** in the main text to improve clarity.

---

> ### Author Response · Authors · 2025-11-20
> **Rebuttal by Authors (3/3)**
>
> ### **W5: The ablation studies are insufficient to isolate design contributions. Table 3 reports limited variations, but it does not disentangle the effects of Fourier filtering, activation injection depth, and steering magnitude, so the relative importance of each component remains unclear.**
>
> In fact, our analysis of Fourier filtering components is already included in **Table 3**, where we separately ablate the information-enhanced (low-frequency) and task-specific (high-frequency) components derived through Fourier filtering.
> The relevant results are as follows:
>
> | Dataset  | ARC-C | FPB | MedMCQA |
> | -------- | --------  | --------  | -------- |
> | **FAST**     | **76.14**     | **59.26**     | **56.09**    |
> | w/o Information-enhanced Activation  | 72.25  | 48.32 | 51.89        |
> | w/o Task-specific Activation | 75.48  | 56.59       | 57.71        |
> | w/o Both                     | 71.80  | 37.80       | 49.40        |
>
> These results show that both components are essential, as removing either leads to performance degradation.
>
> Regarding activation injection depth and steering magnitude, we provide detailed analyses in **Section 4.4** and **Figures 8** (additional results in **Figure 11**).
> We observe that activation steering consistently improves over zero-shot performance across various hyperparameter settings, demonstrating the **robustness** of our approach.
>
> Specifically, injecting activations at middle layers yields the best results, suggesting these layers encode richer features beneficial for cross-task transfer.
> Moreover, our method achieves optimal performance with a steering strength of 0.2. Higher values tend to disrupt the model's original representations, while lower values provide insufficient guidance for the target task.
>
>
>
> ### **Q1: How sensitive is FAST to the choice of frequency cutoff when separating low- and high-frequency components, and how should practitioners select this threshold for new tasks or models?**
>
> In this paper, we set the frequency cutoff threshold $k$ to $d/2$.
> To evaluate the sensitivity of this choice, we conducted experiments with $k$ set to $d/4$, $d/3$, $d/2$, and $2d/3$.
> We use all source tasks in our study and evaluate on ARC-Challenge and Financial Phrasebank as target tasks.
> The results are as follows:
>
> | Frequency cutoff | $d/4$ | $d/3$ | $d/2$ | $2d/3$ |
> | -------- | -------- | -------- | -------- | -------- |
> | ARC-Challenge            | 73.95 | 75.54 | **76.14** | 75.23 |
> | Financial Phrasebank     | 57.33 | 58.57 | **59.26** | 58.12 |
>
> We find that all tested values of $k$ lead to performance improvements over the baseline, demonstrating the **robustness** of our approach.
> Among them, $k=d/2$ yields the **best performance**, as it effectively separates low- and high-frequency components.
>
> We have included these results in **Appendix D.1** to assist practitioners in selecting this threshold for new tasks

---

> ### Author Response · Authors · 2025-11-27
> **Kindly request your feedback before the end of the discussion period**
>
> Dear Reviewer 8p6w:
>
> As the author-reviewer discussion period is soon ending, we would appreciate it if you could review our responses and provide your feedback at your earliest convenience. If there are any further questions or comments, we will do our best to address them before the discussion period ends.
>
> Thank you very much for your time and efforts!
>
> Sincerely,
>
> Authors

---

### Official Review · Reviewer_h5sS · 2025-11-11

**Soundness:** 2
**Presentation:** 3
**Contribution:** 2
**Rating:** 4
**Confidence:** 2

**Summary:**

This paper propose a method for cross-task transfer in prompted language models (which modifies the forward-pass only; no training is involved). The method involves (1) computing the difference vector of the activations on source tasks when given zero-shot prompt vs. few-shot labeled prompts, then (2) injecting this difference vector when solving the target task with zero-shot prompt. The rough idea is that few-shot demonstrations make the model to focus on specialized areas of internal knowledge for solving the task, and we want to induce the same behavior on the target task.

The proposed method has three components:
1. Compute the activation difference vectors on (multiple) source tasks.
    - The authors discussed a graph-based method (section 3.2) for selecting a small subset of the "most useful" examples for the few-shot demonstration.
2. Factor the difference vector into low-frequency and high-frequency components (computed per-layer).
    - They found the high-frequency component to capture more-specialized task-specific activations, hence may not be helpful for transfer when the tasks are dissimilar.
3. Inject the difference vector into an intermediate layer when solving the target task.

**Strengths:**

1. The reviewer is not up-to-date on LLM steering methods, but the proposed method nicely combines few-shot learning, which is very helpful for IID performance but not transfer performance, with activation steering, which is more abstract but could be more "generalizable" as it operates in the latent semantic space.
2. The proposed method is presented in a way that is easy to follow and understand, and intuitive.

**Weaknesses:**

1. Based on discussions in Section 2, the premise of the method is that the difference vector is similar on similar tasks, hence injecting dv to the target task would simulate injecting few-shot demonstration from the target task.  So in order to determine whether the method is applicable in a given scenario, we need to know if the source and target tasks are similar.  For this, the authors (kind of) used task similarity in eq. (3) as a proxy for task similarity.
    - It is unclear from the experiments, that besides from relying on domain knowledge and validation accuracy, whether there's a way to determine if the proposed method would succeed or fail.  The reviewer would have liked to see some failure cases to better understand the limitations of the proposed method: *If I apply this method to unrelated tasks, what will happen?*
    - For example, the authors could evaluate all pairs of the tasks in table 1.
2. The discussion in Section 2.2 is very confusing.  The message seems to be that the change in activation induced by adding few-shot demonstrations is **linearly transferable** (i.e., "parallel") across tasks.
    - But (1) t-SNE is non-linear, so the reviewer is not sure where the conclusion on the differences being parallel comes from.
3. The section on subset selection is a substantial part of the paper (section 3.2), but it appears highly heuristic and the reviewer is unsure of the motivation behind some of its design.
    - The "influence source" seems to favor examples that are clustered together; why is this a desirable objective?

**Questions:**

See Weaknesses.

1. Could the authors evaluate the few-shot demonstration selection method on the few-shot baselines?
2. From evaluating the examples from these tasks, pages 19 and 20, a thing that stood out is that the answer formats are different (labels are ABCD, True/False, or positive/negative/neutral). The reviewer wonders if the few-shot/PEFT baselines can be improved if the answer labels can be made more compatible?

---

> ### Author Response · Authors · 2025-11-20
> **Rebuttal by Authors (1/3)**
>
> We sincerely thank you for the time and effort you have dedicated to reviewing our work.
> We truly appreciate your valuable comments and have carefully addressed all the concerns raised.
>
> In the updated PDF, we have **highlighted the revisions in yellow** for your convenience.
> We hope these modifications adequately address your questions.
>
> ### **W1: Based on discussions in Section 2, the premise of the method is that the difference vector is similar on similar tasks, hence injecting dv to the target task would simulate injecting few-shot demonstration from the target task. So in order to determine whether the method is applicable in a given scenario, we need to know if the source and target tasks are similar. For this, the authors (kind of) used task similarity in eq. (3) as a proxy for task similarity.**
> + **It is unclear from the experiments, that besides from relying on domain knowledge and validation accuracy, whether there's a way to determine if the proposed method would succeed or fail. The reviewer would have liked to see some failure cases to better understand the limitations of the proposed method: If I apply this method to unrelated tasks, what will happen?**
> + **For example, the authors could evaluate all pairs of the tasks in table 1.**
>
> We agree that it is crucial to understand the practical application of our method.
> To systematically analyze the relationship between **task similarity** and **transfer performance**, we conduct experiments evaluating all combinations of representative source tasks (ARC-Challenge, Financial Phrasebank) and target tasks (AG-news, ARC-E, SST2).
> Specifically, we measure task similarity and the transfer effects of different frequency components of the difference vectors.
> Our experimental results are as follows:
>
> + **Target Task: ARC-Challenge**
>
> | Source Task | AG-news | ARC-E | SST2 |
> | -------- | -------- | -------- | -------- |
> | Task Similarity (Different Vector Similarity) | 0.39     | 0.88     | 0.51     |
> | Low-pass Filtered Task Similarity           | 0.75     | 0.78     | 0.73     |
> | High-pass Filtered Task Similarity          | 0.07     | 0.75     | 0.14     |
> | Zero-shot Accuracy                          | 71.80    | 71.80    | 71.80    |
> | + Difference Vectors                        | 71.20    | 75.60    | 72.00    |
> | + Low-pass Filtered Difference Vectors      | 74.80    | 76.20    | 75.40    |
> | + High-pass Filtered Difference Vectors     | 67.60    | 75.40    | 66.80    |
>
> + **Target Task: Financial Phrasebank**
>
> | Source Task | AG-news | ARC-E | SST2 |
> | -------- | -------- | -------- | -------- |
> | Task Similarity (Different Vector Similarity) | 0.68     | 0.38     | 0.84     |
> | Low-pass Filtered Task Similarity           | 0.72     | 0.70     | 0.74     |
> | High-pass Filtered Task Similarity          | 0.66     | 0.05     | 0.76     |
> | Zero-shot Accuracy                          | 37.80    | 37.80    | 37.80    |
> | + Difference Vectors                        | 53.60    | 45.60    | 55.60    |
> | + Low-pass Filtered Difference Vectors      | 55.40    | 54.60    | 55.40    |
> | + High-pass Filtered Difference Vectors     | 51.20    | 32.20    | 56.00    |
>
> From these results, we observe:
> + The effectiveness of directly injecting difference vectors **strongly correlates** with task similarity. When tasks are similar, performance improves; when they are dissimilar, gains are small or even negative.
> + We further decompose the difference vectors via the Fourier transform into low- and high-frequency components. We find that low-frequency components are **more similar** across tasks, while high-frequency components **vary significantly** (see **Section 2.4**).
> + More importantly, the similarity of **low-frequency components serves as a more reliable indicator for transfer**. Low-frequency similarity remains high across tasks, and injecting only low-frequency components consistently improves performance. This suggests that low-frequency features encode general, task-agnostic enhancements, while high-frequency components contain task-specific information that may cause negative transfer (see **Section 2.4**).
>
> We have added a orange bar in **Figure 4** to show the results of directly injecting the difference vector, with explanations in **Section 2.4** in the revised version.

---

> ### Author Response · Authors · 2025-11-20
> **Rebuttal by Authors (2/3)**
>
> Additionally, we include case studies to analyze why directly injecting difference vectors can hurt performance:
>
> + In a small number of cases, when source and target tasks are highly dissimilar, the difference vector introduces source-specific patterns that interfere with reasoning on the target task, causing the model to **output source-task labels**.
>
> ```
> AG-news -> ARC-Challenge:
>
> Input: Given a question answering task from the 3rd to 9th-grade science exam. The question contains four options "A.", "B.", "C." and "D." Select the most appropriate choice that answers the question. Which resource is renewable?\nA. oil\nB. coal\nC. natural gas\nD. water
>
> Ground Truth: D
> Output: Sport (X)
> ```
>
> + In most error cases, the difference vector causes the model to fail on examples originally answered correctly, indicating that high-frequency information from unrelated tasks can **disrupt the model’s prior knowledge**.
>
> ```
> SST2 -> Financial Phrasebank:
>
> Input: The studies are expected to start in 2008.
>
> Ground Truth: neutral
> Output: negative (X)
> ```
>
> ### **W2: The discussion in Section 2.2 is very confusing. The message seems to be that the change in activation induced by adding few-shot demonstrations is linearly transferable (i.e., "parallel") across tasks.**
> + **But (1) t-SNE is non-linear, so the reviewer is not sure where the conclusion on the differences being parallel comes from.**
>
> We would like to clarify the confusion caused by our discussion in Section 2.2.
> When we refer to "parallel" differences, we are describing a **qualitative observation in the 2D t-SNE visualization**: the direction from zero-shot to few-shot activation centroids appears roughly parallel across different tasks **in this low-dimensional space in Figure 1**.
>
> We clarify that this observation is based on a nonlinear projection in the **low-dimensional space** and **does not imply linear parallelism in the original high-dimensional space**.
> In fact, as we further discuss in **Section 2.3**, the alignment of difference vectors in high-dimensional space is closely related to task similarity.
>
> To improve rigor, we have added the statement "in the low-dimensional space" to **Section 2.2**.
>
>
> ### **W3: The section on subset selection is a substantial part of the paper (section 3.2), but it appears highly heuristic and the reviewer is unsure of the motivation behind some of its design.**
> + **The "influence source" seems to favor examples that are clustered together; why is this a desirable objective?**
>
> The motivation for subset selection is that source tasks often contain abundant labeled data, and it is **unnecessary and inefficient** to use all examples for representation extraction and activation steering.
> Our goal is to select a **small subset of samples** that are both **representative and diverse**, enabling efficient activation steering.
>
> The influence score is designed to **identify representative samples** within the task—**not simply to select examples that are clustered together**.
> The effectiveness of this metric has been established in prior subset selection work [1, 2].
> In our method, influence is measured by how well a sample propagates information across the subset.
> High-influence samples typically reach diverse regions of the feature space, rather than being tightly clustered, which can better **capture core task patterns**.
>
> Additionally, our ablation study in **Table 3** shows that removing the influence score leads to performance degradation, confirming that the influence score is essential for subset selection.
>
> [1] Selective Annotation Makes Language Models Better Few-Shot Learners, ICLR 2023.
>
> [2] IDEAL: Influence-Driven Selective Annotations Empower In-Context Learners in Large Language Models, ICLR 2024.

---

> ### Author Response · Authors · 2025-11-20
> **Rebuttal by Authors (3/3)**
>
> ### **Q1: Could the authors evaluate the few-shot demonstration selection method on the few-shot baselines?**
>
> Actually, the few-shot baselines the reviewer mentioned has already been included in our experiments.
> As shown in **Tables 1 and 2** (with full results in **Tables 7 and 8**), we reported results using Few-shot TopK and DPP for demonstration selection.
> These experiments reveal that the effectiveness of cross-task few-shot prompting is highly dependent on the similarity between source and target domains. It helps when tasks are similar, but can introduce noise and hurt performance when they are dissimilar.
>
> As the reviewer suggested, we have now added **two additional common few-shot baselines** that use PPL [1] and Information Score [2] for demonstration selection.
> These results consistently show that even with more advanced selection strategies, cross-task few-shot prompting offers **limited gains**.
> In contrast, our activation steering method brings significantly larger improvements without relying on input expansion.
>
> | Method      | ARC-C | FPB | MedMCQA | SciQ | Social-i-QA | Average |
> | ------      | ------|-----| ------- | ---- | ----------- | ------- |
> | Zero-shot         | 71.80 | 37.80  | 49.40 | 84.40 | 55.40 | 59.76 |
> | Few-shot Random   | 69.31 | 48.51  | 48.69 | 84.63 | 59.14 | 62.06 |
> | Few-shot TopK     | 69.54 | 48.63  | 48.69 | 84.86 | 59.37 | 62.22 |
> | Few-shot DPP      | 69.74 | 48.89  | 49.74 | 85.26 | 59.60 | 62.65 |
> | Few-shot PPL      | 69.64 | 48.72  | 49.45 | 85.07 | 59.44 | 62.46 |
> | Few-shot Information Score    | 69.85 | 49.22	| 49.85 | 85.18 | 59.59 | 62.73 |
> | **Ours**          | **76.14** | **59.26**  | **56.09** | **90.43** | **64.57** | **69.30** |
>
> [1] Demystifying prompts in language models via perplexity estimation, EMNLP 2023.
>
> [2] Finding supporting examples for in-context learning, EMNLP 2023.
>
> ### **Q2: From evaluating the examples from these tasks, pages 19 and 20, a thing that stood out is that the answer formats are different (labels are ABCD, True/False, or positive/negative/neutral). The reviewer wonders if the few-shot/PEFT baselines can be improved if the answer labels can be made more compatible?**
>
> Thank you for your thoughtful comment.
> To address the reviewer's question, we conduct two additional experiments:
> + **Format Alignment with Random Labels**: We unify the answer formats across tasks and use randomly assigned labels.
> + **Format Alignment with Correct Labels**: We unify the answer formats and use correct labels annotated by DeepSeek-R1.
>
>
> | Method      | ARC-C | FPB | MedMCQA | SciQ | Social-i-QA | Average |
> | ------      | ------|-----| ------- | ---- | ----------- | ------- |
> | Zero-shot         | 71.80 | 37.80  | 49.40 | 84.40 | 55.40 | 59.76 |
> | Few-shot Random   | 69.31 | 48.51  | 48.69 | 84.63 | 59.14 | 62.06 |
> | Few-shot TopK     | 69.54 | 48.63  | 48.69 | 84.86 | 59.37 | 62.22 |
> | Few-shot DPP      | 69.74 | 48.89  | 49.74 | 85.26 | 59.60 | 62.65 |
> | Same Format (Random Label)   |68.33|39.25|47.13|84.40|54.20|58.66|
> | Same Format (Corrrect Label) |70.53|51.33|50.76|86.03|60.47|63.82|
> | **Ours**          | **76.14** | **59.26**  | **56.09** | **90.43** | **64.57** | **69.30** |
>
> We observe that even with aligned formats, using incorrect labels misleads the model about the target task, resulting in performance worse than zero-shot method.
> Using correct labels with aligned formats brings an average improvement of 1.3% over the original few-shot methods, confirming that **format mismatch does weaken few-shot performance**.
> However, this approach requires **additional labeling resources** and still underperforms our activation steering method.
>
> This is because aligning formats solves only **surface problems**, it doesn’t resolve the **underlying semantic gaps** between tasks, which may misalign meanings and introduce context noise.

---

> ### Author Response · Authors · 2025-11-27
> **Kindly request your feedback before the end of the discussion period**
>
> Dear Reviewer h5sS:
>
> As the author-reviewer discussion period is soon ending, we would appreciate it if you could review our responses and provide your feedback at your earliest convenience. If there are any further questions or comments, we will do our best to address them before the discussion period ends.
>
> Thank you very much for your time and efforts!
>
> Sincerely,
>
> Authors

---

### Author Response · Authors · 2025-11-24
**General Response**

Dear Area Chairs and Reviewers,

We sincerely thank you for your time, effort, and constructive feedback on our work.

Our paper presents a detailed analysis of **activation patterns** across tasks, revealing that low-frequency components encode task-agnostic, information-enhanced features, while high-frequency components capture task-specific details. Based on this finding, we propose a **Fourier-based Activation Steering** framework for cross-task transfer.

We are truly encouraged that the reviewers acknowledged the contributions of our work, which we briefly summarize below:

+ **Novelty**: Reviewers found our interpretation **compelling and insightful**: low-frequency components correspond to task-agnostic information, while high-frequency components encode task-specific details (**``Reviewers 8p6w``**). They also highlighted that our Fourier-based approach enables cross-task transfer without fine-tuning or retraining, offering a **conceptually elegant and computationally lightweight** framework (**``Reviewer gpcn``**).
+ **Presentation**: Our method is presented in a way that is **easy to follow and understand, and intuitive** (**``Reviewer h5sS``**).
+ **Experiments**: The empirical analysis provides strong evidence supporting the **interpretability** of our approach, and experiments on cross-domain and cross-lingual tasks demonstrate its **broad applicability and robustness** (**``Reviewer 8p6w``**).
+ **Practical Usage**: Our approach requires no architectural changes or additional supervision, enhancing its **practical utility** (**``Reviewer 8p6w``**).

We appreciate the reviewers’ insightful comments and have provided additional experimental results, clarifications, and revisions in the manuscript during the rebuttal phase.
All revisions have been highlighted in the updated manuscript:

**``Reviewer h5sS``**: the corresponding revisions are highlighted in **light yellow**.

**``Reviewer 8p6w``**: the corresponding revisions are highlighted in **light green**.

**``Reviewer gpcn``**: the corresponding revisions are highlighted in **light blue**.

Below we summarize our main responses and revisions:

+ **Clarified Motivation**: We have further explained the motivation and importance of our subset selection and the incorporation of influence scores into the selection process (See [**W3** to **``Reviewer h5sS``**](https://openreview.net/forum?id=HFZhYvKM8V&noteId=xDF1ZD76Yo)). We also emphasized the significance of our empirical analysis for the research community. (See [**W1** to **``Reviewer 8p6w``**](https://openreview.net/forum?id=HFZhYvKM8V&noteId=B9uwfCUN0S)).
+ **Improved writing**: We have revised **Section 2.2** to enhance clarity and rigor (**``Reviewer h5sS``**)
+ **Additional Experiments**:
    + Added experiments on directly injecting difference vectors in **Section 2.4** (**``Reviewer h5sS``**).
    + Moved efficiency analysis from Appendix to **Section 4.4** (**``Reviewer 8p6w``**).
    + Added more ablation stuidies on subset selection in **Section 4.3** (**``Reviewer 8p6w``**).
    + Moved evaluation on generation tasks from Appendix to **Section 4.5** and extended evaluation on multi-modal tasks in **Appendix D.2** (**``Reviewer 8p6w``**)
    + Conducted additional hyperparameter anlysis in **Appendix D.1** (**``Reviewers gpcn and 8p6w``**).
    + Explored more sophisticated signal processing methods (See [**W1** to **``Reviewer gpcn``**](https://openreview.net/forum?id=HFZhYvKM8V&noteId=PxwwrocynK)).
    + Conducted few-shot baselines in [**Q1 and Q2** to **``Reviewer h5sS``**](https://openreview.net/forum?id=HFZhYvKM8V&noteId=visbmtnc5T).

We believe these responses and revisions have significantly improved the technical rigor, clarity, and completeness of our work, and we have incorporated them into the revised version.

Please do feel free to let us know if you have any further questions. We are fully committed to addressing them and welcome the opportunity for further discussion.

Best Regards,

Authors

---

### Author Response · Authors · 2025-11-29
**Summary of Our Responses and Revisions during the Rebuttal Period**

Dear Area Chairs,

We sincerely thank you for your time on effort on our work.

Our paper presents a detailed analysis of **activation patterns** across tasks, revealing that low-frequency components encode task-agnostic, information-enhanced features, while high-frequency components capture task-specific details. Based on this finding, we propose a **Fourier-based Activation Steering** framework for cross-task transfer.

---

## **1. Reviewer Feedback**

During the rebuttal period, we carefully addressed the reviewers' comments through detailed responses and revisions:
+ We regret that **``Reviewers h5sS and 8p6w``** were unable to see our detailed responses and revisions, and we were unable to engage in further discussion with them.
+ **``Reviewer gpcn``** **appreciated our additional results** and acknowledged the **main motivation and novel insights** of our work. However, they maintained their score due to the complexity of our multi-step method. We respectfully believe that our analysis offers valuable insights with the potential to inspire and guide future research. Moreover, we consider our method to be practical and accessible for real-world applications. Please refer to [our detailed response to Reviewer gpcn](https://openreview.net/forum?id=HFZhYvKM8V&noteId=3fa6Vomk6i) for more information.

---

## **2. Our Strengths**

We are greatly encouraged that the reviewers acknowledged the contributions of our work, which we briefly summarize below:

+ **Novelty**: Reviewers found our interpretation **compelling and insightful**: low-frequency components correspond to task-agnostic information, while high-frequency components encode task-specific details (**``Reviewers 8p6w``**). They also highlighted that our Fourier-based approach enables cross-task transfer without fine-tuning or retraining, offering a **conceptually elegant and computationally lightweight** framework (**``Reviewer gpcn``**).
+ **Presentation**: Our method is presented in a way that is **easy to follow and understand, and intuitive** (**``Reviewer h5sS``**).
+ **Experiments**: The empirical analysis provides strong evidence supporting the **interpretability** of our approach, and experiments on cross-domain and cross-lingual tasks demonstrate its **broad applicability and robustness** (**``Reviewer 8p6w``**).
+ **Practical Usage**: Our approach requires no architectural changes or additional supervision, enhancing its **practical utility** (**``Reviewer 8p6w``**).

---

## **3. Main Responses and Revisions**

We appreciate the reviewers’ insightful comments and have provided additional experimental results, clarifications, and revisions in the manuscript during the rebuttal phase.
All revisions have been **highlighted** in the updated manuscript:
Below we summarize our main responses and revisions:

+ **Clarified Motivation**: We have further explained the motivation and importance of our subset selection and the incorporation of influence scores into the selection process (See [**W3** to **``Reviewer h5sS``**](https://openreview.net/forum?id=HFZhYvKM8V&noteId=xDF1ZD76Yo)). We also emphasized the significance of our empirical analysis for the research community. (See [**W1** to **``Reviewer 8p6w``**](https://openreview.net/forum?id=HFZhYvKM8V&noteId=B9uwfCUN0S)).
+ **Improved writing**: We have revised **Section 2.2** to enhance clarity and rigor (**``Reviewer h5sS``**)
+ **Additional Experiments**:
    + Added experiments on directly injecting difference vectors in **Section 2.4** (**``Reviewer h5sS``**).
    + Moved efficiency analysis from Appendix to **Section 4.4** (**``Reviewer 8p6w``**).
    + Added more ablation stuidies on subset selection in **Section 4.3** (**``Reviewer 8p6w``**).
    + Moved evaluation on generation tasks from Appendix to **Section 4.5** and extended evaluation on multi-modal tasks in **Appendix D.2** (**``Reviewer 8p6w``**)
    + Conducted additional hyperparameter anlysis in **Appendix D.1** (**``Reviewers gpcn and 8p6w``**).
    + Explored more sophisticated signal processing methods (See [**W1** to **``Reviewer gpcn``**](https://openreview.net/forum?id=HFZhYvKM8V&noteId=PxwwrocynK)).
    + Conducted few-shot baselines in [**Q1 and Q2** to **``Reviewer h5sS``**](https://openreview.net/forum?id=HFZhYvKM8V&noteId=visbmtnc5T).

---

We believe these responses and revisions have significantly improved the technical rigor, clarity, and completeness of our work, and we have incorporated them into the revised version.

Thank you once again for your valuable time and consideration.

Best Regards,

Authors

---

### Note · Authors · 2026-01-26

I have read and agree with the venue's withdrawal policy on behalf of myself and my co-authors.

---

### Meta-Review · Area_Chair_8RrV · 2025-12-28

**Summary:**

Most reviewers regarded FAST as a novel idea, but found the overall approach overly complex and insufficiently justified. Reviewer h5sS characterized the subset selection component as highly heuristic and lacking clear motivation and ablations. Reviewer 8p6w similarly noted missing ablation studies to disentangle the contributions of Fourier filtering, activation injection depth, and steering magnitude. Reviewer gpcn raised a broader concern that the method is overcomplicated and depends on multiple hard-to-tune parameters, including the frequency cutoff, injection strength, and graph-based sampling hyperparameters. In my view, the rebuttal does not adequately address these methodological concerns, so I recommend rejection.

**Reviewer Concerns:**

I do not think the rebuttal resolves the reviewers’ main concern that the method is overly convoluted. In addition, Reviewer **h5sS**’s concern on Section 2.2 is not addressed by the reviewer (T-SNE is indeed non-linear). The rebuttal also does not answer their broader question of whether there is any principled way to anticipate when the proposed approach will succeed or fail. The additional experiments around AG-news, ARC-E, SST2, ARC-Challenge, and Financial Phrasebank are not sufficient to answer the question, as this set is limited in diversity and does not reflect real-world LLM use cases.

On the positive side, the revised manuscript does address reviewer **8p6w**’s concerns about computational overhead and the narrow evaluation scope, particularly in Sections 4.4 and 4.5.

**Reviewer Scores:**

I think Reviewer h5sS and Reviewer gpcn will keep the score since their concerns are not addressed. Reviewer 8p6w may lower the score to 4 since the reviewer also expressed concern on the method and lack of ablation study, which are not well addressed by the rebuttal.

---

### Decision · Program_Chairs · 2026-01-26

Reject